# Transcriptomic diversity in human medullary thymic epithelial cells

Jason A. Carter[1,2,3], Léonie Strömich [4,11], Matthew Peacey [5], Sarah R. Chapin [1], Lars Velten [6,7], Lars M. Steinmetz [8,9,10], Benedikt Brors [4], Sheena Pinto[4] & Hannah V. Meyer [1✉]

The induction of central T cell tolerance in the thymus depends on the presentation of peripheral self-epitopes by medullary thymic epithelial cells (mTECs). This promiscuous gene expression (pGE) drives mTEC transcriptomic diversity, with non-canonical transcript initiation, alternative splicing, and expression of endogenous retroelements (EREs) representing important but incompletely understood contributors. Here we map the expression of genome-wide transcripts in immature and mature human mTECs using high-throughput 5' cap and RNA sequencing. Both mTEC populations show high splicing entropy, potentially driven by the expression of peripheral splicing factors. During mTEC maturation, rates of global transcript mis-initiation increase and EREs enriched in long terminal repeat retro-transposons are up-regulated, the latter often found in proximity to differentially expressed genes. As a resource, we provide an interactive public interface for exploring mTEC transcriptomic diversity. Our findings therefore help construct a map of transcriptomic diversity in the healthy human thymus and may ultimately facilitate the identification of those epitopes which contribute to autoimmunity and immune recognition of tumor antigens.

[1] Simons Center for Quantitative Biology, Cold Spring Harbor Laboratory, Cold Spring Harbor, NY, USA. [2] Medical Scientist Training Program, Stony Brook University, Stony Brook, NY, USA. [3] Department of Surgery, University of Washington, Seattle, WA, USA. [4] German Cancer Research Center, Heidelberg, Germany. [5] School of Biological Sciences, Cold Spring Harbor Laboratory, Cold Spring Harbor, NY, USA. [6] Centre for Genomic Regulation, The Barcelona Institute of Science and Technology, Barcelona, Spain. [7] Universitat Pompeu Fabra (UPF), Barcelona, Spain. [8] European Molecular Biology Laboratory, Genome Biology Unit, Heidelberg, Germany. [9] Department of Genetics, Stanford University School of Medicine, Stanford, CA, USA. [10] Stanford Genome Technology Center, Palo Alto, CA, USA. [11] Present address: Imperial College London, London, UK. ✉email: hmeyer@cshl.edu

The random generation of T cell receptors during T cell development in the thymus ensures adequate repertoire diversity while presenting a unique challenge for the development of self-tolerance[1]. A key player in the induction of central tolerance are medullary thymic epithelial cells (mTECs), which act as a molecular mirror of the peripheral self through promiscuous gene expression (pGE). This process of pGE allows mTECs to express a comprehensive library of self-antigens[2,3] against which developing T cells are screened and, if reactive, eliminated. Even the absence of a single tissue-restricted antigen (TRA) is sufficient to induce organ-specific autoimmunity in mice[4,5] and human mTEC TRA expression has been shown to correlate with the risk of autoimmune disease[6–8]. While the degree of pGE has been shown to increase with mTEC maturation, the underlying regulatory mechanisms remain poorly understood[9], particularly in the human thymus.

At the core of thymic pGE is the unique transcription factor autoimmune regulator (AIRE)[10,11]. In contrast to conventional motif-specific transcription factors, AIRE appears to recognize generic epigenetic markers and release stalled RNA polymerase II[12–14]. AIRE localizes to super enhancers in mice[15] and works in conjunction with the chromatin remodelers CHD4[16] and BRG1[17]. Consequently, AIRE-induced TRA expression is independent of canonical tissue-specific transcription factors[18–21]. AIRE is not solely responsible for pGE[22], however, with forebrain embryonic zinc-finger-like protein 2 (FEZF2) recently identified as an AIRE-independent regulator of TRA expression[23]. AIRE and FEZF2 together account for approximately 60% of pGE in mice[23].

While mouse mTECs express more than 89% of protein-coding genes[24], the detection of a given gene does not necessarily imply the expression of all possible epitopes contained within that gene. Targeted transcription start site (TSS) analysis in mouse mTECs revealed alternative initiation sites for a subset of TRAs, notably including insulin (*Ins2*) with potential implications for the pathogenesis of type 1 diabetes[18]. Mis-initiation of the *melanoma-associated antigen recognized by T cells* (*MART-1*) gene in human mTECs resulted in the expression of a truncated transcript, impairing central tolerance and consequently generating a measurable antigen-specific T cell response[25]. As AIRE induces TRA expression in a manner independent from typical peripheral regulators[18–21], it is not surprising that mTECs could use different TSSs. Beyond limited examples, the genome-wide prevalence of transcript mis-initiation has not been systematically examined in human mTECs.

Co- and post-transcriptional modifications further increase the number of unique peripheral epitopes[26]. Alternative splicing allows for a single gene to produce multiple unique transcripts with an estimated 10–15% of these transcripts lacking mouse mTEC expression[27] and potentially contributing to autoimmunity[28–30]. Conversely to initial findings of AIRE-mediated promiscuous RNA splicing in mice[31], more recent reports indicate that AIRE-induced genes have lower rates of alternative splicing compared to AIRE-neutral genes[32], with mTECs favoring the production of known splicing variants through a subset of peripheral splicing factors[33]. The extent of alternative splicing has not yet been studied in human mTECs.

Another potential source of epitope diversity is the expression of endogenous retroelements (EREs). Due to their mutagenic potential, EREs are largely silenced in somatic cells; however, domesticated EREs that serve host functions and the occurrence of somatic transposition events suggest that silencing is incomplete[34]. Accordingly, EREs are heterogeneously expressed across diverse human tissues and contribute to the immunopeptidome. In human mTECs, ERE expression is unusual in magnitude and diversity[35]. The significance of this elevated

expression diversity and how ERE expression compares at different mTEC maturation stages are unknown. While mouse models have been invaluable for our understanding of pGE and mTEC biology, a clear picture of the epitope-generating processes in the human mTEC compartment was lacking.

We here employ high-throughput 5'Cap and RNA sequencing to map the expression of genome-wide transcripts in human immature and mature mTECs. Our results demonstrate increased rates of global transcript mis-initiation among the mature mTEC population with transcript initiation stochasticity prevalent among AIRE, but not FEZF2, induced genes. We further show that human mTECs express an average of 20,426 genes, favoring the expression of, on average, 78,573 known transcripts that appear to be driven primarily by differential expression of peripheral splicing factors rather than generating novel transcripts through promiscuous splicing. Finally, specific EREs enriched in long terminal repeat (LTR) retrotransposons are upregulated during mTEC maturation, but contribution to initiation of TRA transcripts is rare, suggesting that ERE expression is largely a passive consequence of pGE. Together, our findings provide molecular insight into the roles of transcript mis-initiation, alternative splicing, and ERE expression in generating human mTEC diversity.

## Results

**Identification of high-quality mTEC transcription start sites.** We first set out to identify TSSs in human mTECs. Through their interaction with developing T cells, mTECs undergo a maturation and differentiation process in which the level of pGE and the total number of expressed TRAs increases (Fig. 1A). The maturation process can be broadly traced on the cellular level by MHCII expression, with *AIRE* and *FEZF2* expression predominately confined to a subset of the mature MHCII^hi mTEC population[23,36–38]. We sorted epithelial cells from five pediatric human thymus samples (Supplementary Table 1) into mature MHCII^hi and immature MHCII^lo mTEC populations (Supplementary Fig. 1A, hereafter referred to as mTEC^hi and mTEC^lo, respectively). To examine genome-wide transcript initiation in mTECs, we used high-throughput 5'Cap sequencing[39–41] which selects for RNA molecules with a 5'Cap followed by standard Illumina sequencing (Fig. 1B). We identified and quantified TSSs at single-nucleotide resolution by aligning the 5'Cap-derived reads and counting the number of transcripts beginning at each genomic position (Fig. 1C), followed by normalizing to tags per million (power-law normalization[42,43], Fig. 1D). As closely clustered single-nucleotide TSSs functionally represent the same start site for a given transcript[44], we collapsed closely related TSSs into transcription start regions (TSRs) and report the set of TSRs meeting a conservative expression threshold. We identified a total of 52,033 unique TSRs, representing 16,117 unique genes (13,322 protein-coding), combined across all mTEC^hi and mTEC^lo samples.

We next assessed the quality of these TSRs. Raw tag counts of TSSs identified using HeliScope Cap Analysis of Gene Expression (hCAGE) single-molecule sequencing follow a power-law distribution[42,43], an observation consistent with our 5'Cap sequencing data and well-conserved across all of our biological replicates (Fig. 1D). Similarly, we found that our 5'Cap sequencing method identified TSRs with an enrichment in TATA motifs and CG nucleotide content that is characteristic of known promoter regions and closely resembles those previously reported in the RefTSS[45] database (Fig. 1E and Supplementary Fig. 2A). We further observed conservation of TSRs across human mTEC samples, with 15% (3466) found in all ten samples and up to 47% (23,109) of TSRs observed in at least two independent samples

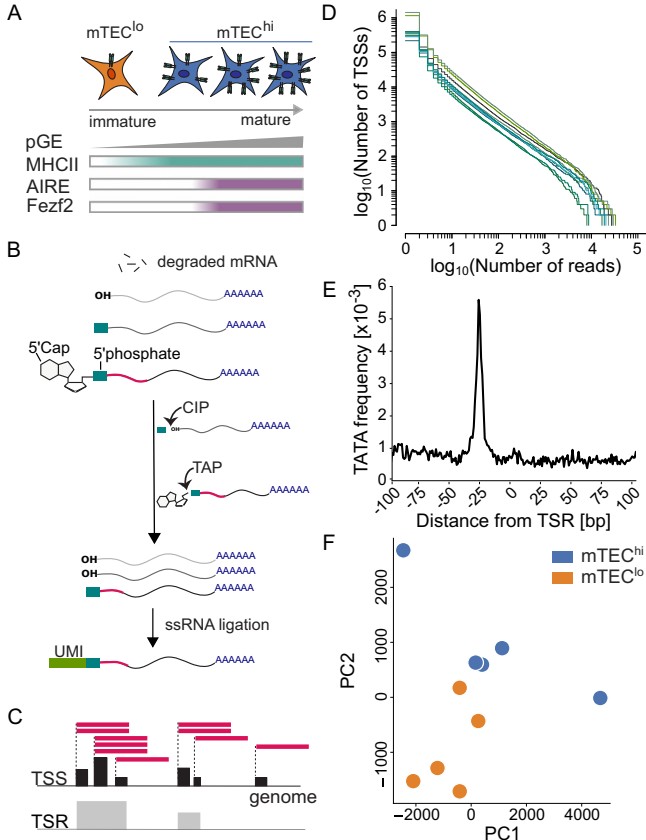

**Fig. 1 Overview of mTEC transcription start regions identified by 5'Cap sequencing. A** Medullary thymic epithelial cells (mTECs) undergo a maturation process that is characterized by the levels of promiscuous gene expression (pGE), MHCII expression, and expression of transcription factors AIRE and FEZF2. **B** Schematic overview of the 5'Cap sequencing method. As previously described[39–41], transcripts with a 5'Cap were isolated using calf intestinal phosphatase (CIP) and tobacco acid pyrophosphatase (TAP) prior to single-stranded RNA (ssRNA) ligation, incorporation of unique molecular identifiers (UMIs), and further library preparation. **C** Following next-generation sequencing, reads were aligned and the number of transcription start sites (TSSs) detected using 5'Cap sequencing were counted at each genomic position. Nearby TSSs were clustered into transcription start regions (TSRs). Distribution of TSSs identified using 5'Cap sequencing followed **D** a power-law distribution (colors represent samples) and **E** enrichment of TATA motifs for TSRs were concordant with those identified in the RefTSS database[45]. **F** Principal component (PC) analysis shows clustering of mTEC$^{hi}$ ($n = 5$) and mTEC$^{lo}$ samples ($n = 5$). Source data for **E** and **F** are provided in the Source Data file.

(Supplementary Fig. 2B). We additionally confirmed that the majority of our TSRs were associated with protein-coding genes and mapped to known promoter regions (Supplementary Fig. 2C–E). Together, our data demonstrate that 5'Cap sequencing allows for the high-throughput identification of genome-wide mTEC TSRs in a reliable and reproducible manner.

**mTEC$^{hi}$ have increased rates of transcript mis-initiation.** TSS usage has been shown to differ between mTECs and their respective peripheral tissues for a limited number of transcripts[18,25]. Given the non-specific mechanisms used by AIRE to induce TRA expression[12–16], we hypothesized that AIRE expression could result in increased transcript initiation stochasticity.

To investigate this, we first needed to understand TSR usage patterns at the global and sample levels. Globally, principal component analysis on normalized TSR expression levels across mTECs showed a clear separation of the mTEC$^{hi}$ and mTEC$^{lo}$ populations (Fig. 1F). At the sample level, all against all correlation analysis showed that the correlation between TSR expression was greater among mTEC$^{hi}$-mTEC$^{hi}$ and mTEC$^{lo}$-mTEC$^{lo}$ samples taken from different individuals than the correlation between mTEC$^{hi}$-mTEC$^{lo}$ pairs taken from the same individual (Supplementary Fig. 3). As these findings demonstrate that TSR patterns within mTEC populations outweigh sample-specific TSRs, we next sought a population-specific TSR usage cut-off which would allow us to study TSR usage differences in these two populations on transcript level. We defined mTEC$^{hi}$-specific TSRs as those 1855 unique TSRs representing 1500 unique genes that were independently observed in at least two mTEC$^{hi}$ samples and not observed in any mTEC$^{lo}$ sample (see Methods and Note: population-specific TSR definition). Using the same definition, we identified 2374 TSRs from 2011 unique genes that were unique to the mTEC$^{lo}$ population.

To assess TSR usage in TRAs, we next calculated a conservative set of human TRAs on both the gene and transcript levels (Eq. (1), Supplementary method: Estimating TRAs from GTEx; Supplementary Data 1 and 2). We identified a total of 7063 TRA-genes and 35,497 TRA-transcripts by applying a previously developed[46] tissue exclusivity score to samples representing 22 human tissue types from the Genotype-Tissue Expression (GTEx) project[47]. As expected given the increased pGE of TRAs in the mTEC$^{hi}$ population (Figs. 1A and 4E), we found that TSRs mapping to TRA-genes were enriched ($p = 2.3 \times 10^{-3}$, by two-sided paired $t$-test) in the mTEC$^{hi}$ population while mTEC$^{lo}$-specific TSRs were more commonly associated ($p = 4.3 \times 10^{-2}$) with housekeeping genes[48] (Fig. 2A).

We then analyzed TSR usage with respect to their dependence on the known transcriptional regulators of pGE, AIRE, and FEZF2 (see Note: human AIRE- and FEZF2-dependent genes). Again as expected given the increased expression of AIRE and FEZF2 in mature mTECs, we confirmed that TSRs associated with AIRE ($p = 4.9 \times 10^{-3}$ by paired $t$-test) and FEZF2 ($p = 2.1 \times 10^{-2}$) dependent genes composed a significantly higher fraction of the mTEC$^{hi}$-specific TSR population relative to the mTEC$^{lo}$-specific TSRs (Fig. 2B). Overall, the FEZF2-dependent-TRA-genes (odds ratio [OR] 15, $p = 1 \times 10^{-6}$ by Fisher's exact test), AIRE-dependent-TRA-genes (OR 2.5, $p = 1 \times 10^{-6}$), and other (i.e., not dependent on either FEZF2 or AIRE) TRA-genes (OR 1.7, $p = 2 \times 10^{-4}$) were significantly enriched in the set of mTEC$^{hi}$-specific TSRs. Housekeeping genes (OR 0.8, $p = 8.2 \times 10^{-2}$) were more commonly found in the mTEC$^{lo}$-specific TSR population (Fig. 2C). Consistent with an increase in the frequency of mis-initiation events, we observed that the mTEC$^{hi}$ population had a statistically significant increase in the number of TSRs identified per gene relative to the mTEC$^{lo}$ population (1.24 vs. 1.17 TSRs per gene, $p = 0.02$ by two-sided Mann–Whitney $U$ test).

We next annotated each TSR according to its genomic location with HOMER[49], finding that mTEC$^{lo}$-specific TSRs were significantly more likely to map to known promoter regions ($p = 4 \times 10^{-5}$ by two-sided paired $t$-test, Fig. 2D). Examination of the distribution of TSRs to non-promoter sites revealed that mTEC$^{lo}$-specific TSRs more frequently mapped to the 5' UTR ($p = 1 \times 10^{-2}$), which are immediately downstream of known promoters (Fig. 2E). The mTEC$^{hi}$-specific TSRs were more commonly found in all other annotated genomic regions, including exonic ($p = 0.001$ by two-sided paired $t$-test), intronic, intergenic, transcript termination sites (all $p \leq 0.01$), and 3' UTRs ($p \leq 0.05$). Non-coding regions were also skewed toward mTEC$^{hi}$

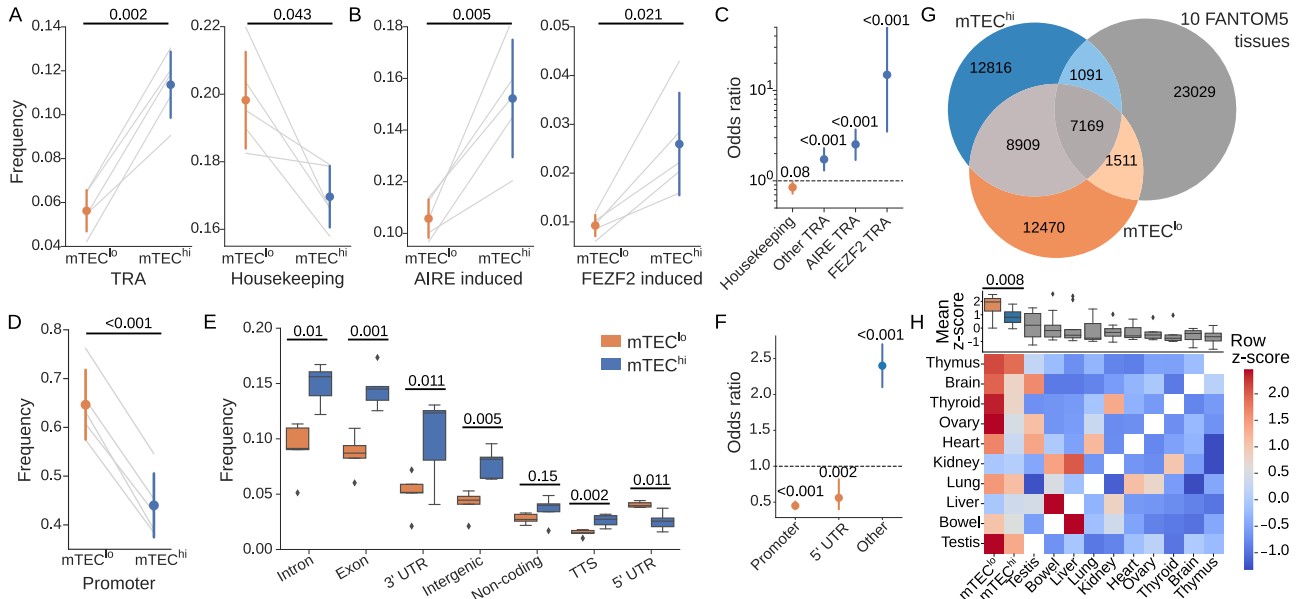

**Fig. 2 The mTEC^hi population has increased rates of transcript mis-initiation.** Comparison of the frequency with which transcription start regions (TSRs) are associated with **A** tissue-restricted antigens (TRAs), housekeeping genes, **B** autoimmune regulator (AIRE)- and FEZF2-dependent genes between paired (gray lines) human mTEC^lo (orange) and mTEC^hi (blue) samples. **C** mTEC^hi vs. mTEC^lo odds ratio comparing the frequency of FEZF2- and AIRE-induced TRAs, other TRAs not associated with FEZF2 or AIRE, and housekeeping genes. **D** Fraction of mTEC-population-specific TSRs mapping to known promoter regions. **E** Distribution of genomic location annotations for TSRs unique to either the mTEC^hi or mTEC^lo populations. Boxplots show median values with interquartile ranges and extrema (whiskers at 1.5× IQR). Outliers beyond 1.5× IQR are shown as dots. TTS transcription termination site, UTR untranslated region. **F** mTEC^hi vs. mTEC^lo odds ratio for TSRs falling into 5' UTR, known promoter regions, and all other annotated regions. **G** Overlap between TSRs found in at least one mTEC^hi, mTEC^lo, and/or normal FANTOM5 peripheral tissue sample[50]. **H** Leave-one-out analysis[51] comparing TSR usage across mTEC^hi, mTEC^lo, and 10 pooled peripheral tissue samples from the FANTOM5 consortium. In brief, one sample type was excluded and the set of TSRs unique to one tissue type among the remaining samples was calculated. The fraction of TSRs expressed in the excluded sample was then reported for each set of otherwise tissue-specific TSRs as a row z-score. The top barplot shows the column mean z-score across all tissue types. Error bars show either standard deviation (**A**, **B**, **D**) or 95% confidence intervals (**C**, **F**). Odds ratio values (**C**, **F**) greater than 1 represent increased use in mTEC^hi TSRs; bars color-coded by population with increased odds. Values above horizontal bars indicate p values derived by two-sided paired t-test (**A**, **B**, **D**, **E**, **H**) or two-sided Fisher's exact test (**C**, **F**); all panels: n = 5 paired mTEC samples. Source data for all panels are provided in the Source Data file.

TSRs ($p = 0.15$). Overall, mTEC^hi TSRs were more than twice as likely to fall outside of these known promoters and nearby 5' UTR regions (OR 2.4, $p = 1 \times 10^{-43}$ by two-sided Fisher's exact test, Fig. 2F). These findings demonstrate a substantial increase in the frequency of mis-initiation events (i.e., those falling outside of known promoter regions) in the mTEC^hi population.

This observed shift away from known promoter regions in the mTEC^hi population is consistent with an increased rate of genome-wide transcript mis-initiation, presumably mediated by non-canonical regulatory mechanisms. We hypothesized that this increase in transcript initiation stochasticity would lead mTECs^hi to share peripheral tissue-specific TSRs less often than mTEC^lo cells. To test this, we compared our mTEC TSRs to TSRs in 10 pooled human organ samples from the FANTOM5 repository[50]. For comparability of TSRs, we downloaded raw FANTOM5 hCAGE datasets and ran the sequencing reads through our TSR calling pipeline (Fig. 1C; see Note: TSR calling approaches). We found that approximately 27% (8268) of mTEC^hi and 29% (8680) of mTEC^lo TSRs were also found in at least one of these 10 FANTOM5 samples, with 37% (16,078) of our unique mTEC TSRs found in both of our mTEC^hi and mTEC^lo populations (Fig. 2G), though these estimates are potentially limited by the comparative lack of sequencing saturation in our mTEC samples (Supplementary Fig. 1B). Performing a leave-one-out analysis as previously described[51], we indeed found that TSRs were more commonly shared ($p = 0.008$ by two-sided paired t-test) uniquely between the mTEC^lo samples and the peripheral tissues (Fig. 2H). While these findings together demonstrate an mTEC^hi-specific

increase in the rate of transcript initiation outside of the known promoter regions used by peripheral tissues, the regulatory roles of AIRE and FEZF2 in this mis-initiation remain unclear.

**AIRE, but not FEZF2, may promote transcript mis-initiation.** We next explored possible regulatory mechanisms responsible for the observed increase in mTEC^hi transcript initiation stochasticity. We first searched for known transcription factor binding motifs within 200 base pairs of each TSR and calculated the mTEC^hi:mTEC^lo OR for each of these known motifs (Fig. 3A). Half of the transcription factor binding motifs found to be enriched around mTEC^lo TSRs belonged to the Krüppel-like factor/Specificity protein (KLF/Sp) family, representing highly-conserved zinc-finger transcription factors that regulate the expression of a diverse set of genes involved with tissue-specific differentiation and development[52]. We found two NF-κB binding motifs to be enriched around the mTEC^hi-specific TSRs (Fig. 3A), consistent with previous findings demonstrating a role for NF-κB activation in inducing mTEC AIRE expression[53,54]. Other promoter motifs enriched in the mTEC^hi TSRs included several transcription factors associated with stem cell pluripotency, including NANOG[55], ASCL2[56], and PRDM10[57]. RNA sequencing performed on three of these paired mTEC^hi and mTEC^lo samples (see Differential TRA expression in human mTECs) showed that these transcription factors are expressed in both populations (Supplementary Fig. 4), however, for the majority of the transcription factors without a significant ($q \leq 0.05$) enrichment in

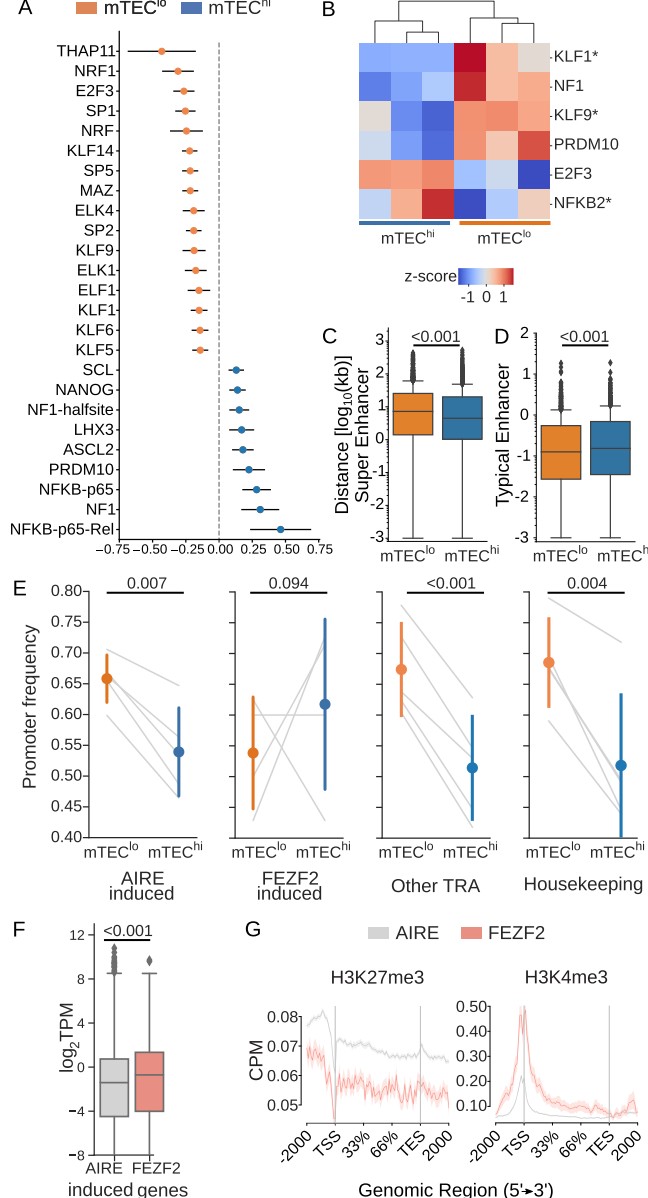

**Fig. 3 AIRE may predominately contribute to transcript mis-initiation.**
**A** Log$_2$ odds ratio (OR) point estimate for transcription factor motifs enriched ($p \leq 0.05$ by Fisher's exact test after Bonferroni correction) within 200 bp of either mTEC$^{hi}$- or mTEC$^{lo}$-specific TSRs. OR values greater than 0 represent increased use surrounding mTEC$^{hi}$-specific TSRs; color-coded by population with increased odds. **B** Of those known transcription factor motifs enriched around mTEC$^{hi}$- or mTEC$^{lo}$-specific TSRs (**A**), only six genes were found to be differentially expressed (Wald test, Benjamini–Hochberg adjusted $q \leq 0.05$) by RNA sequencing. Only 3 of these 6 transcripts (indicated by an asterisk) were enriched in the mTEC population expected from the motif enrichment around the TSRs.
**C** Distance in kilobases (kb) to nearest known super enhancer or **D** typical enhancer in SEdb[59]. **E** mTEC$^{hi}$-specific TSRs were more commonly located outside of known promoter regions for AIRE-induced genes, but not for FEZF2-induced genes. TRAs not induced by either AIRE or FEZF2 and housekeeping genes also show mTEC$^{hi}$-specific TSRs enriched outside promoters. **F** Distribution of transcript-level expression values in mTEC$^{hi}$ cells demonstrates a higher average expression of FEZF2-induced genes relative to AIRE-induced genes. TPM transcripts per million. **G** Mouse Chromatin ImmunoPrecipitation sequencing (ChIP-Seq) data ($n = 4$)[24] demonstrates higher mean (central line) counts per million mapped reads (CPM) of H3K4me3 (histone marker of transcriptional activation, top) for FEZF2 genes. Conversely, higher CPM of H3K27me3 (transcriptional repression, bottom) was observed for AIRE-induced genes. Error bars or bands show either 95% confidence intervals (**A**, **G**) or standard deviation (**E**); boxplots show median (central line) with interquartile range (IQR, box) and extrema (whiskers at 1.5× IQR). Outliers beyond 1.5× IQR are shown as dots (**C**, **D**, **G**). Values above horizontal bars indicate $p$ values derived by the two-sided Mann–Whitney $U$ test (**C**, **D**, **F**) or by the two-sided paired $t$-test (**E**). **A**–**F**: $n = 5$ paired mTEC samples. Source data for **A**–**F** are provided in the Source Data file.

their expression levels (Fig. 3B); i.e., our data demonstrate enrichment of transcription factor sequence motifs around mTEC TSRs without corresponding differential expression of the transcription factor expression (Fig. 3A, B).

We therefore hypothesized that differences in underlying epigenetic regulation between the two mTEC populations could alter the accessibility of these binding motifs and account for the observed sequence enrichment. AIRE localizes to chromatin regions with highly concentrated regulatory elements known as super-enhancers (SEs) in mice, inducing distal gene expression with the help of the chromatin remodeler CHD4[15,16]. While the mTEC chromatin landscape has been explored in mice[58], functional genomic data and thereby TEC-specific SEs are currently not available for human mTECs. As a baseline to investigate the contribution of SEs to transcript mis-initiation, we used the human super-enhancer database (SEdb; representing SEs across 240 human tissue and cell types[59]), and calculated the genomic distance between each TSR and the nearest known SE. We found that the mature mTEC$^{hi}$-specific TSRs were consistently closer to these known SE regions than those TSRs

specific to the mTEC$^{lo}$ population (Fig. 3C). The importance of AIRE-SE mediated interactions in driving mTEC$^{hi}$-specific TSR usage is further supported by the observation that the mTEC$^{hi}$ TSRs were further from typical enhancers than their mTEC$^{lo}$ counterparts (Fig. 3D). Although inter-chromosomal interactions make nucleotide distance a relatively poor proxy for SE function, our data is compatible with mTEC$^{hi}$ specific TSRs more commonly falling within SE neighborhoods and consequently an increased role for non-specific, long-distance interactions mediated by AIRE. We theorized that the use of SE-based long-range regulatory mechanisms employed by AIRE could be the predominant driver of transcript initiation stochasticity in the mTEC$^{hi}$ population. This is in contrast to FEZF2, which has been shown to recognize local promoter sequences in neurons[60] and cooperatively regulate promoter chromatin states in mouse mTECs[16].

Consistent with this hypothesis, we found that AIRE-induced TSRs were more commonly initiated outside of known promoter regions in mTEC$^{hi}$ cells relative to the mTEC$^{lo}$ population ($p = 0.007$ by two-sided paired $t$-test, Fig. 3E and Supplementary Fig. 5A). Importantly, no difference in the frequency of transcript mis-initiation was observed between the two mTEC populations for TSRs associated with FEZF2-induced genes (Fig. 3E and Supplementary Fig. 5B). That is, pGE mediated by AIRE, but not FEZF2, was frequently initiated outside of known promoter regions in human mTECs. To confirm these findings, we calculated the fraction of genes with mis-initiated transcripts (at least one TSR mapping outside of a known promoter region). As the identification of TSRs is dependent on gene expression, we limited this analysis to a set of transcripts with comparable expression levels in our mTEC bulk RNA-seq samples. We indeed found that AIRE, but not FEZF2, induced genes had higher rates

of transcript mis-initiation in the mTEC$^{hi}$ relative to the mTEC$^{lo}$ population (Supplementary Fig. 6A, B).

We further observed that the expression of FEZF2-induced genes was, on average, higher than that of AIRE-induced genes in human mTEC$^{hi}$ cells (Fig. 3F), as previously reported in mouse mTECs[16]. To better delineate the functional role of AIRE in transcript mis-initiation, we next examined epigenetic regulation of AIRE- and FEZF2-induced genes using mouse mTEC Chromatin ImmunoPrecipitation sequencing (ChIP-Seq) datasets[24]. Previous studies have reported increased chromatin accessibility around FEZF2-induced gene TSSs in contrast to increased markers of chromatin repression around AIRE-induced gene TSSs[16]. Extending this analysis across the full gene body, we again found enrichment of the active histone marker trimethylated lysine 4 of histone 3 (H3K4me3) predominantly around the TSS for FEZF2 relative to AIRE-induced genes in the murine mTEC$^{hi}$ population (Fig. 3G). In contrast, examining the trimethylated lysine 27 of histone 3 (H3K27me3) histone marker demonstrated increased epigenetic repression of transcription for AIRE-induced genes across the full gene body relative to FEZF2-induced genes (Fig. 3G). Together, AIRE's localization to distal SEs and recruitment to repressed chromatin markers elevated across the gene body, in conjunction with increased chromatin accessibility for AIRE-induced TRAs at sites distal to the TSS, would suggest a relatively high degree of initiation stochasticity. Under this model, AIRE-induced genes would be expected to more frequently have initiation sites outside of known promoter regions and have overall lower expression, consistent with the findings presented above.

Beyond these AIRE- and FEZF2-induced genes, we additionally found that both housekeeping genes and TRAs not known to be induced by either transcription factor had increased rates of transcript mis-initiation in the mTEC$^{hi}$-specific TSR population relative to the mTEC$^{lo}$ population (Fig. 3F and Supplementary Figs. 5C, D and 6C, D). Limited by the caveat that human AIRE- and FEZF2-induced genes have only been defined by lifting over from the corresponding mouse gene sets, these findings suggest that AIRE is not the sole driver of transcript mis-initiation in mTEC$^{hi}$ and other factors are involved, likely including differences in global chromatin remodeling[61].

In summary, AIRE-induced genes in the mTEC$^{hi}$ population had the highest rates of transcript mis-initiation when compared with AIRE- and FEZF2-independent TRAs and housekeeping genes (Supplementary Fig. 6), strongly supporting a direct role for AIRE, but not FEZF2, in promoting transcript mis-initiation.

**Differential TRA expression in human mTECs**. Previous reports have estimated that mouse mTECs express approximately 89% of known protein-coding genes[24] and extrapolation of gene expression in the complete human TEC compartment (including cortical and medullary TECs) estimates close to 100,000 transcripts[62]. To measure the fraction of gene- and transcript-level TRAs expressed in human mTECs, we performed deep RNA sequencing on paired mTEC$^{hi}$ and mTEC$^{lo}$ samples from three of the five individuals (Supplementary Table 1 and Supplementary Fig. 10). We found that individual human mTEC samples express on average $20,426 \pm 483$ (mean ± standard deviation) unique genes and $78,573 \pm 3664$ unique transcripts, corresponding to an overall average expression of protein-coding genes of $75.6 \pm 0.7\%$. Together, we detected 22,504 (95.9%) protein-coding genes that were expressed at >1 transcript per million (TPM) (see Alternative splicing in mTECs is non-promiscuous) in at least one of the six mTEC samples. In addition, 96.1% of TSRs identified in our previous analyses had corresponding gene expression measurements in this data, with additional expression being detected

for which TSS sequencing yield was likely not deep enough (Supplementary Fig. 1B). Examining differential gene expression between these mTEC populations, we identified 5474 transcripts (3430 unique genes) and 2143 transcripts (1465 unique genes) significantly enriched ($q \le 0.05$, by Wald test) in the mTEC$^{hi}$ and mTEC$^{lo}$ populations, respectively (Fig. 4A).

We first confirmed that expression of the mTEC maturity markers MHCII and CD80 was increased in the mTECs$^{hi}$ (Fig. 4A, B), consistent with the mTEC development pathway (Fig. 1A). Similar to findings in mice, we found that expression of *FEZF2* (125-fold) and *AIRE* (292-fold) was higher in human mTECs$^{hi}$ relative to mTECs$^{lo}$ (Fig. 4C). We further confirmed the enrichment of nine previously reported human mTEC$^{hi}$ marker genes[63] in our dataset (Fig. 4D). These findings suggest a high degree of homology between mouse and human mTEC maturity stages.

We next compared the expression of TRAs across the mTEC$^{hi}$ and mTEC$^{lo}$ populations. We identified 1747 TRA-transcripts (1348 unique TRA-genes) and 374 TRA-transcripts (273 unique TRA-genes) significantly ($q \le 0.05$) enriched in the mTEC$^{hi}$ and mTEC$^{lo}$ population, respectively (Fig. 4E). Consistent with the function of AIRE and FEZF2 in mature mTECs, differentially expressed TRAs were more than twice as likely to be expressed by the mTEC$^{hi}$ population (OR 2.2, $p = 7.7 \times 10^{-39}$ by Fisher's exact test). TRA-genes enriched within the mTEC$^{lo}$ population were strongly associated with gene ontology terms[64] related to the skeletal and cardiac muscle (Fig. 4F). These findings are consistent with single-cell sequencing data suggesting that the expression of muscle-specific antigens was predominantly limited to a cluster of MHCII-low myoid cells in the human thymus[63].

To further assess differences in the coverage of TRAs by mTEC maturity, we calculated the fraction of TRA-genes and TRA-transcripts expressed based on a TPM threshold ($\ge 1$ TPM) in the mTEC$^{hi}$ and mTEC$^{lo}$ populations. We found thymic representation was lowest for adipose and testis TRA-genes, as previously reported in mice[51] (Fig. 4G). TRA-transcripts mapping to adipose tissue, the spinal cord, and frontal cortex showed the lowest levels of thymic representation (Fig. 4H), potentially reflecting a lack of certain peripheral tissue-specific splicing factor expression in the thymus[33]. Comparing TRA coverage by the stage of mTEC maturity, we again found that the mTEC$^{lo}$ population expressed a higher fraction of both genes and transcripts with expression in the periphery restricted to the heart and skeletal muscle (Fig. 4F–H). The mTEC$^{lo}$ population additionally expressed a higher fraction of ovarian and cerebellum TRA genes and transcripts. While a higher fraction of both TRA-genes and TRA-transcripts for all other sampled tissue types were expressed by the mTEC$^{hi}$ population, the difference in expression between the two mTEC populations was more pronounced on TRA-gene than TRA-transcript level (Fig. 4I). That is, the mTEC$^{lo}$ expressed a relatively higher number of TRA-transcripts than TRA-genes when compared to the mTEC$^{hi}$ population. Together, these findings potentially reflect a higher rate of mTEC$^{lo}$ alternative splicing.

**Alternative splicing in mTECs is non-promiscuous**. Mouse mTECs express a significantly higher number of isoforms than are found on average in peripheral tissues[31,51] and seem to generate these known isoforms by reusing peripheral splicing factors[33]. Furthermore, AIRE-induced genes seem to have lower rates of alternative splicing events when compared with AIRE-neutral genes[32]. As our examination of TRA expression demonstrated a relatively increased expression of transcripts per gene in our mTEC$^{lo}$ population, we hypothesized that differential

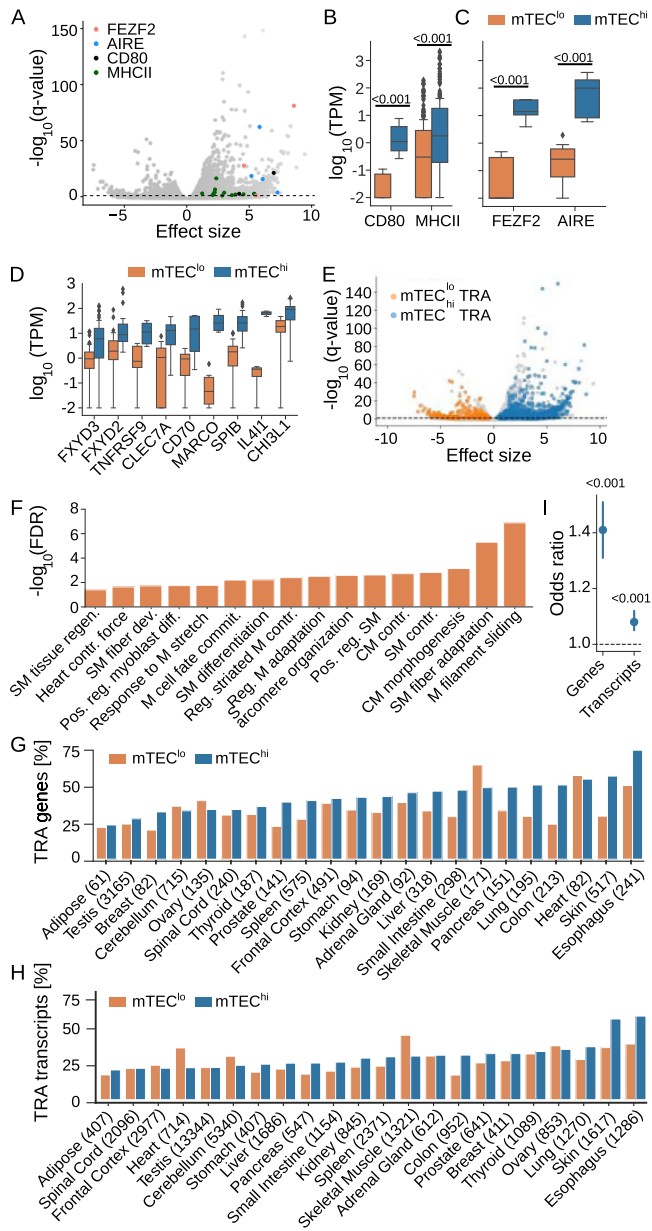

**Fig. 4 Differential expression of tissue-restricted antigens across mTEC populations. A** Enrichment of the mature mTEC marker *CD80* and *MHCII* transcripts, as well as *AIRE* and *FEZF2* transcripts in the mTEC$^{hi}$ population (Wald test, effect size ≥0). **B** Comparison of average expression across all transcripts for *CD80*, *MHCII*, **C** *FEZF2*, and *AIRE*. **D** mTEC$^{hi}$ marker genes identified previously[63] were similarly enriched in our mTEC$^{hi}$ population: FXYD3, FXYD2, TNFRSF9, SP1B ($p < 0.001$), CD70 ($p = 0.01$), MARCO ($p = 0.005$), IL4L1 ($p = 0.1$) and CH13L1 ($p = 0.004$); by two-sided Mann–Whitney $U$ test. **E** Enrichment of 1747 and 374 tissue-restricted antigen (TRA) transcripts in the mTEC$^{hi}$ and mTEC$^{lo}$ populations, respectively (Wald test, effect size ≥0). **F** Gene ontology analysis of those TRAs enriched in the mTEC$^{lo}$ population revealed a strong preference for muscle-related functions. SM skeletal muscle, CM cardiac muscle, M muscle, Regen regeneration, Pos. reg. positive regulation, Contr contraction, diff differentiation, Commit commitment. **G** Percentage of tissue-specific TRA genes and **H** transcripts expressed in the mTEC$^{lo}$ and mTEC$^{hi}$ populations; absolute numbers in parentheses. **I** Odds ratio of mTEC$^{hi}$:mTEC$^{lo}$ TRA expression on gene and transcript level. The relative expression of TRA transcripts was more similar between the two populations. Error bars show 95% confidence intervals. Boxplots show median (central line) with interquartile range (IQR, box) and extrema (whiskers at 1.5× IQR). Outliers beyond 1.5× IQR are shown as dots (**B**, **D**). Values above horizontal bars indicate $p$ values derived by the two-sided Mann–Whitney $U$ test (**B**, **C**) or Fisher's exact test (**I**); all panels: $n = 5$ paired mTEC samples. Source data for all panels are provided in the Source Data file.

transcriptome regardless of the chosen threshold, closely followed by the mTEC$^{lo}$ and mTEC$^{hi}$ populations. The mTEC$^{hi}$ population expressed a larger percentage of the transcriptome only at relatively low thresholds (TPM ≤0.55), with nearly 60% of known transcripts having detectable expression in the mTEC samples. Selecting an expression threshold of 1 TPM, we next modeled the total number of expressed transcripts as a function of expressed genes for all tissue types. We found that while the mTEC$^{lo}$ population expressed a higher number of transcripts per gene, the number of expressed transcripts in both the mTEC$^{hi}$ and mTEC$^{lo}$ samples appeared to be directly related to the increased number of genes expressed in these samples (Fig. 5C). Consistent with recent findings in mice[32], we found that the mTEC$^{hi}$ population expressed a lower than expected number of transcripts per AIRE-dependent gene (Supplementary Fig. 7A). In contrast, no such difference was observed for the number of transcripts expressed per FEZF2-dependent, AIRE- and FEZF2-independent TRA, or housekeeping gene in the mTEC$^{hi}$ population (Supplementary Fig. 7B, D).

Finally, we asked whether the differential expression of peripheral splicing factors could account for differences in mTEC isoform diversity. Using rMATS[65] to quantify differential alternative splicing between stages of human mTEC maturity, we found a total of 3411 differentially expressed isoforms (FDR ≤0.05 and a difference in percent spliced greater than 10%; Fig. 5D). A slight majority (55%) of these were enriched in the mTEC$^{lo}$ population, including a greater number of skipped exons, retained introns, and alternative 5' and 3' splice sites. The mTEC$^{hi}$ population demonstrated a slightly higher number of mutually exclusive exons. When specifically considering AIRE-dependent, FEZF2-dependent, AIRE- and FEZF2-independent TRA, and housekeeping transcripts, alternative splicing patterns were consistent with the whole transcriptome analysis though higher rates of skipped exons were observed for AIRE- and FEZF2-dependent transcripts in the mTEC$^{hi}$ population (Supplementary Fig. 8). Notably, no specific difference in the distribution of alternative splicing events was observed for AIRE-dependent transcripts. To determine whether the preferential expression of

expression of peripheral tissue splicing factors could drive an increase in human mTEC$^{lo}$ alternative splicing.

To investigate alternative splicing in mature and immature human mTECs, we first calculated splicing entropy (Eq. (2)), for our mTEC and 25 GTEx healthy peripheral tissue samples. On a per gene basis, a higher splicing entropy represents more diverse isoform expression (i.e., splicing entropy for a given gene is maximal when all isoforms for a given gene are equally expressed). The overall splicing entropy reported for each tissue is then the sum of all genes with an expression of more than one transcript in that tissue. With the notable exception of the testis, the mTEC samples had significantly higher splicing entropy relative to all other surveyed peripheral tissue types (Fig. 5A). This finding is consistent with previous observations in mice[31] and suggests that the human thymus similarly expresses a large number of isoforms. No significant difference was detected in splicing entropy between the mTEC$^{lo}$ and mTEC$^{hi}$ populations.

We next calculated the fraction of known transcripts expressed in each tissue type at varying TPM thresholds (Fig. 5B). We again found that the testis expressed the highest fraction of the

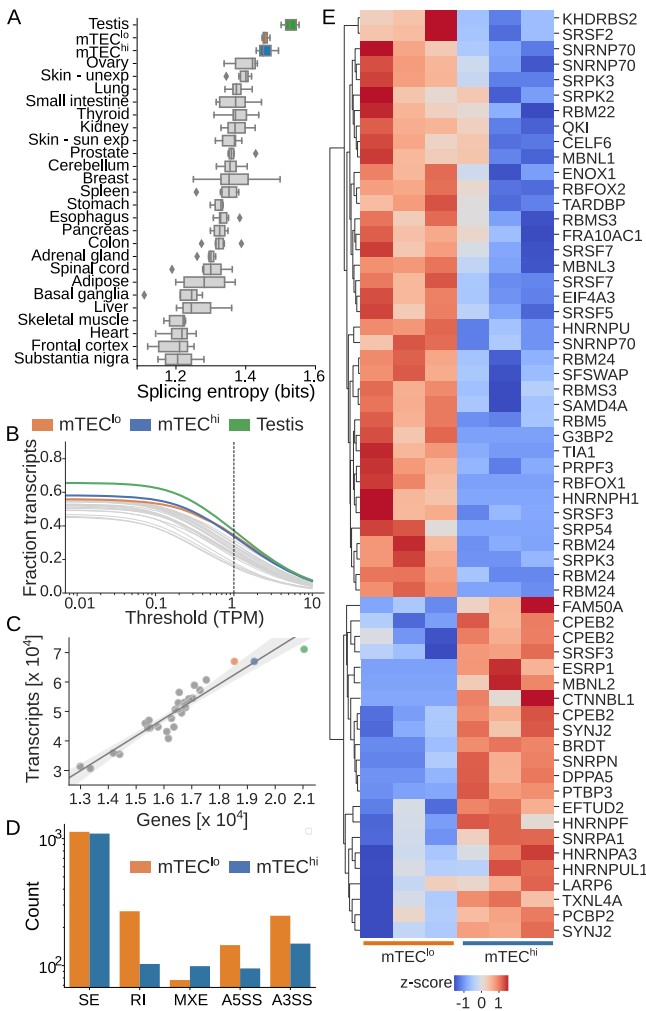

**Fig. 5 Alternative splicing in mTECs is mediated by differential expression of peripheral splicing factors. A** Splicing entropy calculated for mTEC[hi] and mTEC[lo] as well as 25 healthy peripheral tissue samples from GTEx[47]. Boxplots show median (central line) with interquartile range (IQR, box) and extrema (whiskers at 1.5× IQR). Outliers beyond 1.5× IQR are shown as dots. **B** Fraction of transcriptome expressed at varying transcript per million (TPM) thresholds for mTEC and peripheral tissue samples. **C** Linear regression fitting the number of expressed transcripts as a function of the number of expressed genes in healthy GTEx samples predicts number of transcripts in mTEC[hi] and mTEC[lo] samples. The gray shaded area marks the 95% confidence interval. **D** Differential splicing between the mTEC[hi] and mTEC[lo] populations as predicted by rMATS[65] divided by skipped exons (SE), retained introns (RI), alternative 5' and 3' splice sites (A5SS, A3SS), and mutually exclusive exons (MXE). **E** Clustermap showing row z-scored expression of known peripheral tissue splicing factor transcripts significantly ($q \leq 0.05$ by Wald test) enriched in either the mTEC[lo] (count = 38) or mTEC[hi] (count = 22) population. All panels: $n = 5$ paired mTEC samples, $n = 6$ samples per GTEx tissue. Source data for **A**, **C–E** are provided in the Source Data file.

peripheral splicing factors could account for these differences, we searched for differentially expressed splicing factors from previously identified 287 RNA binding proteins[66–68]. We identified 60 unique transcripts representing 48 genes that were differentially expressed ($q \leq 0.05$) between mature and immature mTECs (Fig. 5E, full list of splicing factors and their expression provided in Supplementary Data 3). Together, our findings support the generation of peripheral tissue isoforms through the use of known splicing factors[33].

**LTR retrotransposons are upregulated during mTEC maturation.** ERE expression is unusually elevated in the human mTEC compartment[35]. To investigate if ERE expression changes during mTEC maturation and how it compares to peripheral tissue expression, we quantified the expression of ERE subfamilies from the LTR, SINE, LINE, and composite classes ("Other") in mTEC[hi] and mTEC[lo] populations, as well as embryonic stem cells and the 25 healthy peripheral tissue samples from GTEx. We found that the observed increase in ERE expression magnitude and diversity relative to peripheral tissues has contributions from both mTEC populations (Fig. 6A). In addition, we noticed differences in ERE expression between mTEC[hi] and mTEC[lo] and hypothesized that specific EREs that contribute to antigen diversity could be induced during mTEC maturation. Of 818 subfamilies, differential expression analysis identified 77 upregulated and 49 downregulated subfamilies (Fig. 6B), with upregulated subfamilies enriched for LTR retrotransposons ($p = 2.2 \times 10^{-4}$ by Fisher's exact test; Supplementary Fig. 11A). LTR retrotransposon derived peptides have also been found enriched in EREs that contribute to the immunopeptidome as ERE-derived MHC-I associated peptides (ereMAPs) in B-lymphoblastoid cell lines (B-LCLs)[35]. To profile the expression of these ereMAPs, we quantified the expression of EREs at a locus level and mapped the ereMAP sequences back to their genomic coordinates. Of the 108 ereMAP loci identified in B-LCLs, 100 and 99 ereMAP transcripts were detected (RPKM >1) in mTEC[hi] and mTEC[lo] populations, respectively, and only four ereMAP loci were differentially expressed between mTEC[hi] and mTEC[lo] populations. Overall, ereMAP loci were most highly expressed in mTEC populations but heterogeneously expressed across peripheral tissues (Supplementary Fig. 11B). This suggests that while ereMAP loci are abundantly expressed in mTECs, their expression is largely independent of the maturation state.

Having quantified the expression of ERE subfamilies on a per locus level, we observed that their expression is consistent with the expression profile of genes located within 1000 bp of the EREs' TSS (Fig. 6C); i.e., start sites of ERE loci upregulated in mTEC[hi] cells are enriched in the vicinity of mTEC[hi] upregulated genes, and likewise for downregulated ERE loci and genes ($p < 10^{-16}$ and $p = 7.1 \times 10^{-15}$, respectively, by Fisher's exact test). As co-option of ERE promoters can drive the expression of genes in early development and oncogenes in tumorigenesis[69], we hypothesized that initiation from ERE promoters could contribute to antigen diversity by facilitating pGE. We used LIONS to measure transcription initiated by EREs[70] and detected 61 ERE-initiated chimeric transcription events (Supplementary Data 4), with 18 of those specific to the mTEC[hi] population (Fig. 6D), exemplified by the initiation of the protein-coding gene *LRRC61* from the promoter of LTR *MER41B* (Fig. 6E). ERE-initiated genes included both protein-coding and long non-coding RNAs (lncRNAs), consistent with previous studies reporting EREs as a major source of lncRNA transcripts[71]. However, the protein-coding genes initiated from ERE promoters did not show a preference for TRAs. In summary, while we find that ERE expression is a common feature of mTECs irrespective of maturation stage and yields transcripts that can translate to ereMAPs, there is currently no evidence that ERE expression actively contributes to pGE by initiating transcription independent of AIRE- and FEZF2-driven mechanisms.

## Discussion

Thymic negative selection identifies and removes autoreactive T cells, in part, through the ectopic expression of peripheral self-antigens by mTECs. Through use of genetic models, these processes have been studied in great detail in mouse, but a detailed

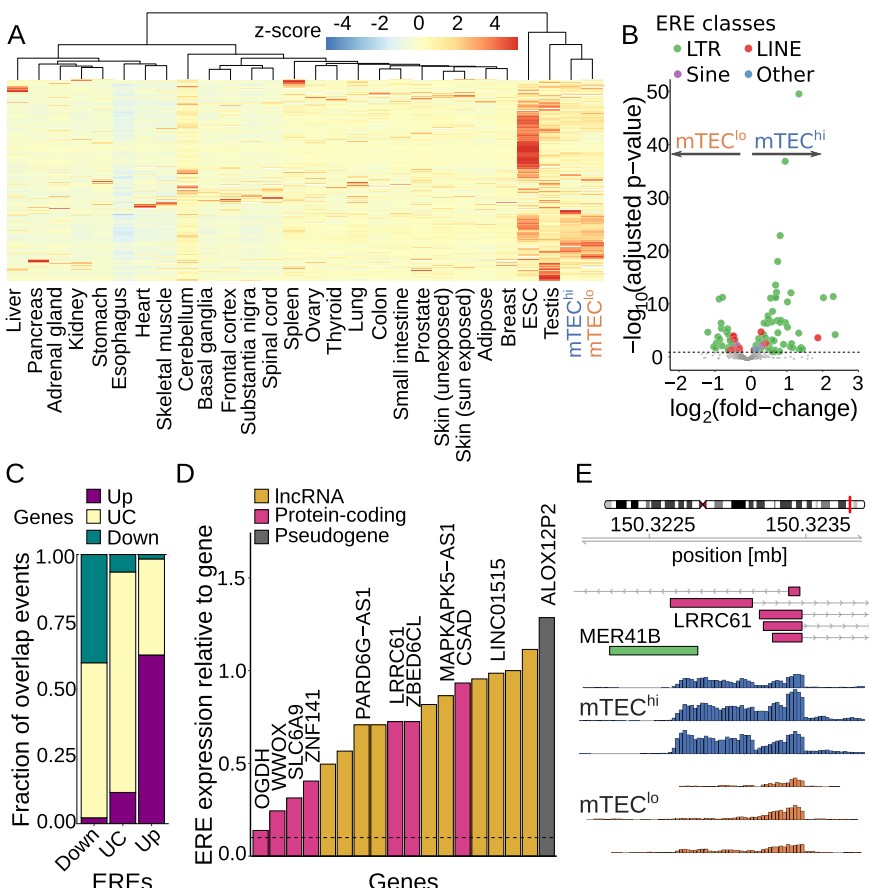

**Fig. 6 Changes in ERE expression during mTEC maturation. A** Expression (in z-scored TPM counts) for 613 ERE subfamilies identified with SalmonTE[108] in mTECs, embryonic stem cells (ESCs), and 25 GTEx tissues. **B** Differential expression (Wald test) of 818 ERE subfamilies detected by TEtranscripts[106] during mTEC maturation, colored by class. **C** Overlap of expression status between genes and EREs with a TSS within the gene body ±1000 bp. The gene expression status is color-coded by the differential expression between mTEC^hi and mTEC^lo: Up upregulated, Down downregulated, UC unchanged; corresponding categorization for ERE expression on x-axis. **D** Contribution of ERE expression to the expression of an annotated gene for each ERE-initiated chimeric transcript unique to the mTEC^hi population detected by LIONS[70]. Transcripts are colored by gene biotype; lncRNA long non-coding RNA. Genes with annotated gene names are highlighted. **E** MTEC^hi-specific initiation event from the *MER41B* promoter (LTR) into the protein-coding gene *LRRC61*. Expression tracks show counts per million in the mTEC populations. All panels: n = 5 paired mTEC samples, n = 6 samples per GTEx tissue. Source data for **A**–**D** are provided in the Source Data file.

examination of the different layers of gene expression regulation in the human thymus has been missing. Here, we combined high-throughput 5'Cap sequencing with standard RNA sequencing to investigate gene expression and transcript diversity in immature and mature human mTECs. We provide an interactive transcriptome browser to explore the transcript diversity we uncovered at http://transcriptomediversity.cshl.edu/.

Targeted TSS analyses in mouse and human mTECs have pointed to mis-initiation events for a limited number of peripheral antigens, leading to the escape of autoreactive T cells into the periphery[18,25]. To investigate transcription initiation events on a genome-wide level, we profiled TSS in five paired mature mTEC^hi and immature mTEC^lo healthy human mTEC samples. We found that mTEC^hi had substantially higher rates of transcript initiation outside of known promoter regions when compared with mTEC^lo. Likely reflecting differences in the underlying regulatory mechanisms, transcript mis-initiation was observed at increased rates for AIRE-dependent, but not FEZF2-dependent, genes in the human mTEC^hi population.

To investigate whether differences in chromatin regulation could underlie AIRE-specific transcript mis-initiation, we reanalyzed mouse ChIP-seq data as we are unaware of any human

mTEC functional genomic data. We found increased markers of chromatin activation around FEZF2-induced genes, in contrast to increased markers of chromatin repression across the full gene body of AIRE-induced genes in mouse mTECs. Similarly, studies in mice have shown increased chromatin accessibility around FEZF2-induced promoters relative to AIRE-induced genes[16]. The high degree of chromatin accessibility, combined with FEZF2's action at proximal promoter sequences[60], would mechanistically favor a low degree of initiation stochasticity and thereby support the induction of relatively high gene expression levels at canonical TSSs. In contrast, AIRE's localization to distal SEs[15] and non-specific recruitment to regions of silenced chromatin[12,24] likely contribute to increased transcript mis-initiation in AIRE-induced genes. Other potential contributors to mTEC^hi transcript mis-initiation include the action of chromatin remodeler BRG1, with less than 10% of mouse BRG1-induced chromatin opening occurring within 1 kb of known TSSs[17]. To directly assess these epigenetic mechanisms in human, further work investigating AIRE, FEZF2, and BRG1 binding in conjunction with chromatin accessibility in human mTECs is needed. In addition to mechanistic insight, those results will also aid in deriving human-specific FEZF2- and AIRE-dependent gene lists.

We next studied overall pGE and alternative splicing in human mTECs. Earlier work on the levels of pGE in human mTECs relied on micro-array analysis and relative comparisons between different subsets of mTECs[72,73], whereas more recent studies provide an atlas of the human thymus focused on general thymus biology with limited work on TEC-specific questions[63,74]. Both approaches are additionally limited in their ability to detect splice variants. We performed RNA sequencing on a subset of samples, detecting 22,504 (95.9%) protein-coding genes with expression >1 TPM in at least one of our six mTEC samples. Muscle-related genes were differentially expressed in the mTEC[lo] population, consistent with the MHCII[lo] myoid cluster identified in the human thymus atlas[63,74]. While mature mTECs[hi] expressed a significantly higher number of TRA genes than immature mTECs[lo], this difference was less pronounced at the TRA transcript level. We detected a corresponding increase in the number of splicing factors differentially expressed in the mTEC[lo] population, similar to results obtained in a recent study in mice[33]. They identified RBFOX splicing factors as influential on TEC development, and our analysis shows the same differential expression of *RBFOX1* in human as in mice, a finding worth future investigation. Overall, however, the total number of expressed transcripts was proportional to the number of expressed genes in both mTEC populations relative to peripheral tissues. Together these findings are consistent with mTECs generating known isoforms through the re-use of peripheral splicing factors[33] rather than AIRE-dependent promiscuous splicing[31].

Larouche et al. showed that the expression of EREs is unusually elevated in the mTEC compartment[35]. Here, we have extended their findings by identifying a specific set of EREs which are induced during mTEC maturation and enriched for LTR retrotransposons. In general, ERE expression mirrors that of nearby genes, but this could not be explained by transcription initiation from EREs. Instead, it is likely that ERE expression is a passive consequence of epigenetic remodeling that enables pGE[20]. Interestingly, mature mTECs exhibit loss of imprinting at the *IGF2* locus[22], suggesting they share epigenetic features with the pre-implantation embryo, in which ERE expression is also widespread[75]. Future functional genomic studies in human mTECs may shed light on the cause of ERE expression by probing the chromatin landscape at ERE loci.

Ultimately, we hope that future studies will be able to expand upon these features of thymic transcriptome diversity and lead to the creation of a comprehensive database of the epitopes responsible for inducing thymic tolerance in humans. By comparing such a genome-wide set of thymic epitopes with those encountered in the periphery, missing epitopes responsible for autoreactive T cell escape or for driving immune recognition of tumor neoantigens could be systematically identified. While the present study represents an important first step, comprehensive sequencing of TSRs, complete isoform coverage, and further studies on regulatory mechanisms for ERE expression in human mTECs are still necessary to achieve this goal.

Specifically, our TSR analysis has shown that 5'Cap sequencing followed by standard Illumina sequencing[39–41] is a powerful tool for genome-wide transcription initiation analysis without relying on highly specialized single-molecule sequencing. However, in comparison to standard RNA-seq in the matched thymic samples or hCAGE data from FANTOM5[50], we find that we have not reached saturation levels for the detection of TSRs in mTECs. Deeper sequencing[76] and additional human mTEC samples will be necessary to refine the definition of mTEC population-specific TSRs and achieve comprehensive coverage for comparison with peripheral TSRs.

On the transcript level, enhanced epitope maps in mTECs could be generated using long-read sequencing for the discovery of potentially novel transcript isoforms. Peptidomic experiments in human mTECs will additionally be necessary to correlate this transcriptomic epitope expression with the peptide diversity directly observed by developing T cells. Furthermore, our understanding of the complexities of mTEC maturation is still evolving and is not fully captured by the canonical mature MHC[hi]- and immature MHC[lo]-populations[36–38]. Careful delineation of these mTEC sub-populations, such as via the lectin Tetragonolobus purpureas agglutinin staining to distinguish post-AIRE MHC[lo] mTECs from immature MHC[lo] mTECs[77], will thereby be an important component of future studies. Finally, studies investigating the specific roles of mTEC pGE in the development of various T cell subsets[78,79], including autoreactive regulatory T cells (Tregs)[80], will also be of interest.

In conclusion, our study has demonstrated several mechanisms that underlie the generation of transcriptomic diversity in human mTECs. Our results represent an important first step toward the generation of a detailed understanding of the mTEC transcriptome and ultimately the identification of epitopes not seen by developing T cells during the induction of central tolerance. Future comprehensive identification of these missing antigens will play a crucial role in the identification of epitopes with the potential to trigger autoimmune responses against healthy tissue or drive immune recognition of tumor neoantigens.

## Methods

**Statistical tests**. Appropriate statistical tests and their *p* value are reported for each analysis. All tests were conducted in a two-sided manner.

**Tissue-specific antigens**. We used the measure $\tau$ as introduced in ref. [46] and benchmarked in ref. [81] to determine a list of human TRAs on both gene and transcript levels (TRA-genes and TRA-transcript, respectively):

$$\tau = \frac{\sum_{i=1}^{n}(1 - \hat{x}_i)}{n - 1} \text{ with } \hat{x}_i = \frac{x_i}{\max_{1 \le i \le n} x_i} \tag{1}$$

where $x_i$ is the expression of the gene/transcript in tissue $i$ and $n$ is the number of tissues. We started with 40,481 genes and 194,360 transcripts across the 22 human tissues obtained from the GTEx consortium (see GTEx peripheral tissue expression data). After filtering for all genes/transcripts with a TPM ≥1, we obtained 26,131 genes and 118,256 transcripts for which we independently computed $\tau$. All genes/transcripts with $\tau \ge 0.8$ were considered as TRAs. To determine the tissues in which these TRAs are expressed, we used the binarization approach also described by Yanai et al.[46]. For each TRA, expression values are sorted and the maximum difference between sorted values is detected. All tissues with expression larger than the value separated by the maximum difference are considered as expressing this TRA. Estimation of TRAs, tissue-specificity of tissue-specific genes, and comparison to previous methods in Supplementary document, section Estimating TRAs from GTex.

**Human Aire- and Fezf2-dependent genes**. While knockout studies in mice have defined lists of FEZF2- and AIRE-dependent genes[23,24], such lists are difficult to experimentally derive for human mTECs. However, murine and human AIRE and FEZF2 protein have high protein sequence similarity (71% and 96%, respectively) and all functional protein domains are conserved[82] (Supplementary Fig. 9). Furthermore, *AIRE* mutations in human patients and *Aire* knockout studies in mice show similar phenotypes with autoreactive antibodies and lymphocyte infiltration in the same tissues[3,83]. Together with the evidence for conserved mechanism guiding gene regulation in mouse and human[84], we followed previous studies[73,85] to obtain AIRE- and FEZF2-dependent gene lists, by using orthologues of murine FEZF2[23] and AIRE-dependent genes[24]. Lists were converted from mouse to human orthologues using *biomaRT* (v2.46.3 [86]), via attribute *hsapiens_homolog_ensembl_gene*. (a) AIRE-dependent genes: AIRE-dependent genes were defined by Sansom et al. as differentially expressed genes between *Aire* knockout mTECs and mature *Aire* expressing mTECs in mice (Benjamini–Hochberg corrected *p* values ≤0.05). Supplementary Table 3, sheet 16 of ref. [24] provides all differentially expressed genes, applying a fold change threshold of ≤2 yields the 3980 AIRE-dependent genes described in the paper. Our homology matched this set of AIRE-dependent genes to human, obtaining a final list of human AIRE-dependent genes that consisted of 3361 unique ensembl gene identifiers. The class of AIRE-dependent TRA genes ($n = 2213$) was established by intersecting AIRE-dependent genes and TRA genes. (b) FEZF2-dependent genes: Hiroyuki Takaba kindly provided a full table of normalized gene expression in wild-type mTECs and *Fezf2* knockout mTECs described in ref. [23]. Genes that were differentially expressed between wild-type and knockout

mTECs ($p$ value ≤0.05 and fold change ≥2 in WT mTECs vs. *Fezf2* KO mTECs) were homology mapped to human. The final list of human FEZF2-dependent genes included 256 unique ensembl gene identifiers. These were merged with the TRA gene list to obtain FEZF2-dependent TRA genes ($n = 195$).

**Human housekeeping genes**. Eisenberg and Levanon published a full list of human housekeeping genes which we mapped from refseq to ensemble gene identifiers, yielding 4211 unique genes[48].

**FEZF2 conservation**. Murine and human FEZF2 protein sequences were obtained from uniprot with the identifiers sp|Q9ESP5 and sp|Q8TBJ5, respectively, and were aligned using clustal omega (v1.2.4 [87]). CD-Search[88] was used to identify their protein domains.

**Human thymus tissue**. Human thymus samples were obtained in the course of corrective cardiac surgery at the Department of Cardiac Surgery, Medical School of the University of Heidelberg, Germany; informed written consent was obtained from all patients; patients were not compensated for taking part in the study. The study was approved by the Institutional Review Board of the University of Heidelberg.

**Tissue processing**. MTEC samples were isolated as described previously[25]. In brief, thymi were digested sequentially with three rounds of collagenase/dispase for 20 min each at 37 °C, followed by trypsin for 10 min each at 37 °C in a water bath with magnetic stirring. The trypsin fractions were pooled and filtered through 60 μm gauze. Enrichment of mTECs was performed by magnetic cell sorting followed by cell staining and FACS. Magnetic cell sorting was performed using anti-CD45 Microbeads (Miltenyi Biotech, Germany). The labeled CD45+ cells were depleted using the autoMACSTM Pro Separator (Miltenyi Biotech). The enriched stromal cell fraction (CD45−) was stained with biotinylated anti-epithelial cell adhesion molecule (EpCAM/sav-PE clone HEA125, kindly provided by Gerhard Moldenhauer, DKFZ, 1:100 dilution), CDR2-Alexa488[89] (cortical dendritic reticulum antigen 2, DKFZ, Alexa Fluor 488 Protein Labeling kit; Molecular Probes, Invitrogen, Germany, 1:100 dilution), Alexa 680-conjugated mAb HLA-DR[73] (Alexa Fluor 680 Protein Labeling kit; Molecular Probes; clone L243, kindly provided by Gerhard Moldenhauer, DKFZ, 1:500 dilution), and anti-CD45- PerCP[73] (clone 2D1, BD Biosciences, 1:100 dilution). MTECs were sorted as CD45-, CDR2-, EPCAM+ cells and MHCII (HLA-DR) was used to separate immature mTEC^lo and mature mTEC^hi cell populations. Dead cells were excluded with propidium iodide (0.2 μg/ml). Cell sorting was performed on a FACS Aria (BD Biosciences). The gating strategy is depicted in Supplementary Fig. 1A (BD FACSDiva Software (v8.2) and FlowJo -(v7.6)). RNA from sorted mTEC^hi and mTEC^lo populations was extracted using the High Pure RNA Isolation Kit (Roche).

**5'Cap sequencing of human mTEC populations**. TSSs were sequenced following the detailed protocol described in[40]. This method pre-selects only those RNA molecules from the RNA lysate that carry a 5'Cap (Fig. 1B), i.e., captures true start sites of transcripts, ignoring the 5' ends of degraded and fragmented RNA strands. Library preparation included the ligation of unique molecular identifiers (UMIs) to ensure PCR amplifications can be identified. Libraries were sequenced on Illumina HiSeq 2000. Reads were demultiplexed, UMIs trimmed via *umi_tools extract* (umi_tools v1.1[90]), screened for contamination with *fastq_screen* (v0.14.0 [91]), aligned to GRCh38/Gencode annotation (release 33) using STAR (v2.7.2b [92]) and deduplicated based on UMIs using *umi_tools dedup*. Deduplicated, aligned reads were filtered for uniquely mapped reads via *samtools view -b -q 255* (samtools v1.11[93]). Quality control of sequencing and alignment was conducted using *FastQC* (v0.11.8 [94]) and *picard CollectRnaSeqMetrics* (v2.18.20 [95]) and summarized with *multiqc* (v1.9 [96]). Throughout the manuscript, we graphically show paired mTEC^lo and mTEC^hi taken from the same individual and report the results of appropriate, two-sided paired-sample statistical testing whenever possible.

**Transcription start regions in human mTEC populations**. For TSS calling, only the forward (i.e., 5' end) of each read pair was retained via *samtools view -h -f 0x40*. TSS was then defined as the 5' position of the uniquely mapped, forward reads. The expression levels of the TSSs for each mTEC sample were normalized to tags per million using the power-law normalization implemented in *CAGEr* (v1.32 [42]). After normalization, *removeBatchEffect* in *limma* (v3.46 [97]) was used to remove sequencing batch effects. Normalized TSSs in each sample were combined into TSRs using *paraclu* (v9 [44]) with a minimum tag cluster expression of 2 tags per million and a maximum cluster length of 20bp. TSRs across samples were combined with *bedops* (v2.4.38 [98]) and consensus, strand-specific TSRs in human mTEC samples called by merging TSRs derived from different samples within 20bp proximity using *bedtools merge* (v2.29.2 [99]). The variability of TSR expression across samples, and specifically the high prevalence of TSRs expressed only in a subset of mTEC samples, made identification of mTEC population-specific TSRs difficult using conventional differential gene expression approaches (see Supplementary Note: Population-specific TSR definition). In order to identify mTEC^hi- and mTEC^lo-specific TSR sequences that were reproducible (i.e., detected in at least

two independent samples), we empirically defined mTEC^hi-specific TSRs as those TSRs that were expressed in at least two mTEC^hi and not detected in any mTEC^lo samples; mTEC^lo-specific TSRs were called accordingly. TSRs were annotated using HOMER (v4.11.1 [100]), including mapping to the closest gene, calculation of TSR CpG/GC content (*-CpG*), and TATA motif search with the provided motif file (*-m HOMER/motifs/tata.motif*).

**Transcription start region motif and enhancer analyses**. Transcription factor motifs were identified within 200 base pairs around each TSR using HOMER[49] (*findMotifsGenome.pl*). We included the 400 curated transcription factor motifs using the default HOMER list for both mTEC^lo- and mTEC^hi-specific TSRs independently. The mTEC^lo:mTEC^hi OR was then calculated according to the number of TSRs in each population containing a given transcription factor motif. Those transcription factors with a Bonferroni-corrected $p$ value ≤0.05 were reported. Super- and typical-enhancer coordinates were downloaded from the human SEdb[59] (http://www.licpathway.net/sedb/), which encompasses a comprehensive set of enhancers drawn from 240 human tissue and cell types. We calculated the linear intra-chromosomal distance in base pairs between each of our TSRs and the nearest super- or typical-enhancer in SEdb.

**FANTOM5 transcription start site analysis**. Raw FANTOM5 hCAGE sequencing data for brain, colon, esophagus, heart, kidney, liver, lung, ovary, small intestine, testis, thymus, and thyroid were downloaded from the DNA Data Bank of Japan at ftp://ftp.ddbj.nig.ac.jp/ddbj_database/dra/fastq/DRA000/; for index and accession numbers see Supplementary Table 2. Reads were aligned to GRCh38/Gencode annotation (release 33) using STAR (v2.7.2b [92]) and filtered for read quality of at least 255 with *samtools view -q 255*. The 5' position of the uniquely mapped, forward reads were defined as TSS, and the TSRs calling pipeline (described for mTEC samples above) was applied to obtain TSRs in the ten tissues analyzed. Analogous to consensus TSRs calling within mTECs samples, we also computed consensus TSRs from mTEC and FANTOM5 TSRs.

**Saturation analysis of TSS**. For each of the ten mTEC samples, the total number of demultiplexed, qc-ed, and aligned 5'Cap reads was determined with *samtools idxstats*. For a specified set of read thresholds downsampling proportions dependent on the samples' total library size were computed and *sambamba view* (v0.8 [101]) used to downsample libraries to the specified proportion. After downsampling, each set of reads was processed as described in 5'Cap sequencing of human mTEC populations. We then counted the number of TSS discovered at each read threshold as the number of 5' position of the uniquely mapped, forward reads.

**RNA-seq of human mTEC populations**. The libraries were prepared manually using the NEBNext®Ultra™ Directional RNA Library Prep Kit according to the manufacturer's instructions. Libraries were sequenced using Illumina HiSeq 2000. Quality control was conducted using *FastQC* (v0.11.8 [94]).

**GTEx peripheral tissue expression data**. Raw bam files including both aligned and unaligned reads for 25 tissues (Adipose Subcutaneous, Adrenal Gland, Brain Basal Ganglia, Brain Cerebellum, Brain Frontal Cortex (BA9), Brain Spinal cord (cervical c1), Breast Mammary Tissue, Colon Transverse, Esophagus Mucosa, Heart Left Ventricle, Kidney Cortex, Liver, Lung, Muscle Skeletal, Ovary, Pancreas, Prostate, Skin Sun Exposed, Skin Not Sun Exposed (Suprapubic), Small Intestine Terminal Ileum, Spleen, Stomach, Substantia Nigra, Testis, Thyroid) were obtained from the GTEx consortium through dbGap (accession: phs000424, authorized access for project #28176, approval 2/24/21; Supplementary Table 3). Skin (Sun Exposed), Basal Ganglia, and Substantia Nigra were not used in the calculation of TRAs in order to avoid duplicate tissue types. For each tissue, we choose three samples from each sex (if applicable) and, when possible, equally from the following age ranges: <30, 30–60, and >60. The resulting bam files were then sorted (*samtools sort*) and converted to fastq files using *bedtools bamtofastq*.

**RNA-sequencing read processing and quantification**. For mTEC and GTEx samples, reads were filtered with *fastp* (v0.11.8 [102]) for minimum phred quality (*-q 25*) of at most 10% unqualified bases (*-u 10*), a minimum length of 50 bp (*-l 50*), at least 30% complexity (*-y*) and polyX tail trimming (*-x*). Filtered reads were pseudo-aligned and quantified using *Kallisto* (v0.4.6 [103]) with 100 bootstrap samples.

**Differential expression analysis**. *Sleuth* (v0.30.0 [104]) was used for differential expression analysis between mTEC^hi and mTEC^lo samples, adjusting for patient effect in the reduced and full model. Transcripts with $q$ value ≤0.05 were considered differentially expressed. Gene annotation was added to transcript IDs using *biomaRt* (v2.46.3 [86]). List of annotated transcripts (Fig. 4A): AIRE transcripts— ENST00000530812, ENST00000527919, ENST00000291582, ENST00000337909, ENST00000397994; FEZF2 transcripts—ENST00000283268, ENST00000475839, ENST00000486811; CD80—ENST00000478182, ENST00000264246; MHCII— ENST00000411959, ENST00000484643, ENST00000468299, ENST00000461508, ENST00000383126, ENST00000437183, ENST00000433975, ENST00000419685,

ENST00000429783, ENST00000546801, ENST00000443184, ENST00000374975, ENST00000449560, ENST00000476192.

**Splicing entropy and abundance analyses**. For mTEC and GTEx samples, filtered reads were aligned to GRCh38/Gencode annotation (release 33) using STAR (v2.7.2b [92]) with the following options: *outFilterType BySJout, outFilterMultimapNmax 1, alignSJoverhangMin 8, alignSJDBoverhangMin 1, outFilterMismatchNmax 999, alignIntronMin 20, alignIntronMax 1000000* and *alignMatesGapMax 1000000*. Splicing entropy was calculated for a given gene as in ref. [31] with:

$$\sum_{i=1}^{n} P(g_i)\log_2(P(g_i)) \qquad (2)$$

where $P(g_i)$ is the relative expression of the *i*th isoform of gene *g*. The overall entropy for a sample is defined as the median splicing entropy for all genes with >1 isoforms and is reported in bits.

**Differential splicing analysis**. *rMATS* (v4.1.1 [65]) was used for differential splicing analysis between mTEChi and mTEClo samples.

**Endogenous retroelement read mapping and quantification**. ESC RNA-seq data from[105] was downloaded from SRA (accessions: SRR488684 and SRR488685). Filtered reads were aligned to GRCh38 using STAR (v2.7.2b [92]) with the following options: *sjdbOverhang 100, winAnchorMultimapNmax 200, outFilterMultimapNmax 100*. Aligned reads were used for raw transposable element/gene counts quantification using *TEtranscripts* (v2.2.1 [106]) or *TElocal* (v1.1.1 [107]) with default parameters, for subfamily- and locus-level quantification, respectively. To obtain TPM counts at a subfamily level, *SalmonTE* (v0.4 [108]) was run on the filtered reads in "quant" mode with the following options: *reference=hs, exprtype=TPM*.

**Endogenous retroelement differential expression analysis**. ERE counts were extracted (RepeatMasker class "LTR", "LINE", "SINE" or "Retroposon"; Retroposons in text referred to as "composite") and filtered for entries with a minimum of 2 reads across samples. Differential expression analysis on this set of counts was performed with DESeq2 (v1.30.1 [109]). EREs with a Benjamini–Hochberg -adjusted *p* value < 0.05 were considered differentially expressed.

**Endogenous retroelement genomic position analysis**. EREs encoding MAPs were identified by intersecting the genomic coordinates of previously identified MAPs[35] with the RepeatMasker annotation. Genes nearby EREs were identified by searching for GENCODE V38-annotated genes within 1000 bp of RepeatMasker-annotated ERE start sites.

**Endogenous retroelement chimeric transcription analysis**. Chimeric transcription events were detected with LIONS[70] using the "oncoexaption" preset and recurrence/specificity settings of 2 and 1, respectively.

**Mouse ChIP-seq analysis**. ChIP-seq data for H3K4me3 and H3K27me3 in mTEChi cells derived from 4-week-old C57BL/6 mice described in ref. [24] was obtained from SRA (SRP033578, runs SRR1045003-SRR1045008) using *fastq-dump*. Raw read files were split into forward and reverse reads using "fastq-dump –split-files" before quality control with *fastp* (v0.11.8 [102]). QCed reads were aligned with STAR (v2.7.2b [92]) using default parameters. Picard tools (v2.18.20 [95]) "picard MarkDuplicates" with "–REMOVE_DUPLICATES true" was used to remove duplicates before merging aligned reads of biological replicates into a single file. *ngs.plot.r* (v2.61 [110]) was used to analyze and visualize the results for H3K4me3 and H3K27me3 ChIP signal across TSS and gene bodies ("ngs.plot.r -R TSS" and "ngs.plot.r -R genebody") of AIRE- and FEZF2-dependent genes (see Gene lists section below).

**Note: population-specific transcription start region definition**. When compared with traditional RNA-seq data, 5'Cap-seq TSR abundance is much more variable across samples[111], Supplementary material. Specifically, even TSRs that are highly expressed in several samples may not be detected in many other samples (Supplementary Fig. 2). This variability across samples increases the statistical noise of TSR expression and makes it difficult to identify differentially expressed TSRs using traditional differential gene expression pipelines such as DESeq2[109] or Sleuth[104].

Consider the following example of a TSR that is expressed in none of the five mTEClo samples (represented by vector [0,0,0,0,0]) and expressed at 100 transcripts per million in $\frac{3}{5}$ mTEChi samples ([100,100,100,0,0]). By paired *t*-test, this yields a *p* value of 0.07, which would not be considered statistically significant at a common $\alpha$ level of $p = 0.05$, even before correcting for multiple testing. As such, only a very small number of TSRs would be called as differentially expressed using standard methods and we felt it necessary to seek alternative definitions that would allow us to empirically define TEC population-specific TSRs. We defined a TSR as specific to either of the mTEC populations if it was expressed in at least two samples from that population and not detected in the other population. This ensures that TEC-specific

TSRs are reproducible (expressed in at least two independent samples) without being too restrictive for downstream analysis.

A further consideration is that our 5'Cap sequencing approach does not allow for absolute saturation and detection of every TSR from each sample (Supplementary Fig. 1B). It is therefore possible that some of the "TEC-specific" TSRs using our approach might be expressed in both populations. However, incorrectly calling TSRs as specific to one of the populations when they are actually shared would make the two populations appear more similar to one another in our subsequent analyses. The observed differences between the mTEClo and mTEChi populations identified using this method are therefore likely conservative.

**Note: transcription start region calling approaches**. In the following section, we will briefly review the original method for TSR calling in the FANTOM5 database and contrast it with our approach.

The FANTOM5 consortium investigated TSSs in 998 human and 394 mouse tissues using single-molecule hCAGE[111]. To analyze these data, they developed a custom method for identifying peaks in the CAGE profiles called decomposition peak identification (DPI; https://github.com/hkawaji/dpi1/; described in ref. [111], Supplementary material). DPI takes aligned and quality-controlled HeliScope reads from each sample, converts them to BigWig files, and subsequently pools the BigWig files from all samples/tissues into one BigWig file which is used to determine consensus TSRs. In brief, any single-nucleotide start site supported by 2 or more reads across all samples was selected and grouped into tag clusters with other nearby start sites (within 20 bp). The resulting tag clusters passing a size length threshold (greater than 50 bp) were then segregated into distinct transcription start events by applying independent component analysis. On these components and the short tag clusters (less than 50 bp), TSRs were called at different stringency thresholds of evidence, yielding a set of permissive and robust consensus TSRs. To obtain sample-specific TSR profiles, the consensus TSRs were overlapped with the TSS per sample where any overlap detected is considered expressed. Counts of TSRs per sample were then normalized to tags per million using the relative log expression method in edgeR[112].

As described in Methods: Transcription start site analyses, we choose a conceptually similar, but a different approach for calling TSR in human mTECs. After normalizing the read counts in each mTEC sample to tags per million using a power-law transformation, we applied *paraclu*[44] on each sample to call TSRs in each sample independently, choosing a maximum length of 20 bp per TSR and a minimum coverage of two tags per million. We then analyzed the sample-specific TSRs for consensus clusters by combining TSRs across samples with *bedops* and calling consensus, strand-specific TSRs by merging TSRs derived from different samples within 20bp proximity using *bedtools merge*.

The main difference between the two approaches lies in the sample pre-processing and the level at which consensus clusters are called. In the FANTOM5 analysis, samples were pooled and consensus clusters called before any individual sample-specific TSR was derived; in addition, thresholding for expression level (tags per million) was applied after TSR calling. Using this approach makes the discovered clusters specific to the samples included in the analysis. In particular start regions with low support (low tags per million) that are found in similar sample types could be considered, while those with similarly low support in underrepresented samples/tissue types would correctly not be considered in the consensus clustering approach. It also makes it more difficult to extend any analysis to samples/tissues that were not included in the original analyses, as the entire calling process would have to be repeated to allow for a fair comparison.

Initially, we attempted to re-run the DPI approach published with FANTOM5, but found the implementation was specific to the original authors' compute system. We re-implemented the pipeline in a compute-system-independent framework with integrated version management (Snakemake[113]), allowing for the analysis to run on any high-performance compute cluster or on any local computer. The re-implementation can be found here: https://github.com/meyer-lab-cshl/dpi1. However, after careful consideration of the drawbacks outlined above, we decided to conduct the comparison between FANTOM5 and mTEC samples by first re-analyzing the FANTOM5 aligned HeliScope reads with our pipeline for TSR calling. In the future, this makes the TSRs for FANTOM5 usable for comparisons with other, previously not investigated tissue or cell types and makes it independent of the subset of tissues analyzed.

**Transcriptome diversity application**. The data from this study is displayed via the Transcriptome Diversity application at http://transcriptomediversity.cshl.edu/. The interactive plots displaying the TSR, 5'Cap sequencing, and RNA-seq data were generated with *Gviz* (v1.34.1 [114]). The interactive heatmaps were generated with *InteractiveComplexHeatmap* (v2.9.4 [115]). The Gviz plots and the interactive heatmaps were deployed as an RShiny application using the *ShinyDashboard* (v0.7.1). Individual aligned and qc-ed read files from different samples were combined via *samtools merge* to generate one track each for the mTEChi and mTEClo populations. The code for the browser application can be found at https://github.com/meyer-lab-cshl/transcriptomic-diversity-human-mTECs-shiny.

**Reporting summary**. Further information on research design is available in the Nature Research Reporting Summary linked to this article.

## Data availability

The 5′ cap and RNA sequencing data generated in this study have been deposited in the GEO database under accession code GSE201720. The processed sequencing data are available at https://github.com/meyer-lab-cshl/transcriptomic-diversity-human-mTECs with corresponding python notebooks to reproduce the figures. The data used to generate figures in this manuscript are provided in the Supplementary information/Source Data file. An interactive interface to explore the data is available at http://transcriptomediversity.cshl. edu/. Additional data used in this study are available at the following sources: ChIP-seq data for H3K4me3 and H3K27me3 in mTEC^hi cells derived from 4-week-old C57BL/6 mice at SRA: SRP033578 [https://0-www-ncbi-nlm-nih-gov.brum.beds.ac.uk/Traces/study/?acc= PRJNA230856&o=acc_s%3Aa] (runs SRR1045003-SRR1045008)[24]; embryonic stem cell RNA-seq data at SRA: SRR488684 and SRR488685[105]; raw gene expression bam files for 25 tissues (Adipose Subcutaneous, Adrenal Gland, Brain Basal Ganglia, Brain Cerebellum, Brain Frontal Cortex (BA9), Brain Spinal cord (cervical c1), Breast Mammary Tissue, Colon Transverse, Esophagus Mucosa, Heart Left Ventricle, Kidney Cortex, Liver, Lung, Muscle Skeletal, Ovary, Pancreas, Prostate, Skin Sun Exposed, Skin Not Sun Exposed (Suprapubic), Small Intestine Terminal Ileum, Spleen, Stomach, Substantia Nigra, Testis, Thyroid) were obtained from the GTEx consortium through dbGap (accession: phs000424). Source Data are provided with this paper.

## Code availability

Custom analysis code was written in either R (version ≥4.0.3) or python (version ≥3.8). The analysis code is freely available on GitHub: https://github.com/meyer-lab-cshl/ transcriptomic-diversity-human-mTECs (https://doi.org/10.5281/zenodo.6648501).The code for the browser application can be found at https://github.com/meyer-lab-cshl/ transcriptomic-diversity-human-mTECs-shiny.

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

## Acknowledgements

The research was supported by the Simons Center for Quantitative Biology at Cold Spring Harbor Laboratory; the Cold Spring Harbor Laboratory and Northwell Health Affiliation; US National Institutes of Health Grant S10OD028632-01 and 1R01AI167862-01; and by the G. Harold & Leila Y. Mathers Foundation. J.A.C was supported by NIHGM MSTP Training award T32-GM008444. M.P. is supported by a George A. & Marjorie H. Anderson Fellowship. We thank M. Gale Hammel for her advice on the ERE analysis, T. Janowitz, D. Fearon, A. Siepel, and M. Gale Hammel for critical feedback on the manuscript, and R. Ramani and A. Barton for setting up the data server. This work was inspired by many discussions with our mentor and colleague, the late Dr Bruno Kyewski.

## Author contributions

J.A.C., L.S., L.V., M.P., and H.V.M. conducted the analyses; S.P. prepared samples and libraries; S.R.C. and H.V.M. implemented the shiny app; L.M.S., B.B., S.P., and H.V.M. conceived the study, managed the project, and provided critical interpretation of the results; J.A.C. and H.V.M. co-wrote the manuscript with help by M.P.; all authors edited and revised the final version of the manuscript.

## Competing interests

The authors declare no competing interests.
