## [Peer Review File · Nature Communications]

REVIEWER COMMENTS

Reviewer #1 (Thymic selection, systems immunology) (Remarks to the Author):

The authors use 5'Cap- and RNA-seq to profile expression profiles in human thymic samples, with cells purified from the mTEClo and mTEChi compartments. Overall, the authors use this dataset to assess PGE and mTEC maturity in a number of ways. It comprises a useful resource to the community thymic immunobiologists but I have a number of reservations with regards to the manuscript's conclusions.

Specific comments

Inconsistent capitalisation of genes/protein names plus PGE vs. pGE.

“with minor correlations between samples taken from the same individual” – could the authors explain more about what this means? How often were the same TSRs observed in replicate samples from the same patient? If this happens infrequently, does that mean that some observed signals could be sequencing noise rather than genuine TSRs?

Is there any evidence for additional specific TEC chromatin modifications at mTEChi/mTEClo TSRs? On a similar note, were the enhancer and super-enhancer definitions used here based on the TEC chromatin landscape (rather than pooled thymus tissue, which would be mostly thymocytes)? This could be lifted over from mouse (e.g. enhancer chromatin modifications from <https://www.frontiersin.org/articles/10.3389/fimmu.2018.02120/full>).

The authors report enrichment for Nfkb, Nanog, Ascl2 and Prdm10 around mTEChi TSRs. There is limited support for these as important in mTEChi vs. mTEClo from RNA-seq. However, there is often a disconnect between transcriptome and proteome for transcription factors: have the authors tried to assess these transcription factors in mTEChi vs. mTEClo by FACS or similar methods?

The definition of mTEChi and mTEClo specific TSRs is based on detection in at least 2 samples of one mTEC subtype and non-detection in the other subtype. Is there a statistical rationale for this? Would this be called as significant by any standard differential analysis approach (e.g. DESeq2)?

Does the analysis account for multiple samples being taken from the same patient? Would further sequencing of these libraries improve overlap between samples (as suggested in Figure S1B)?

“Aire could be the predominate” -> “Aire could be the [predominant]”

Although it looks convincing that TSRs are located outside the promoter region for AIRE-regulated genes, the plot for FEZF2-regulated genes is much less convincing. Given the amount of noise in the FEZF2 Figure 3E plot, can the authors be sure that there isn't a cell type-specific effect, which may be driven by differential relative contribution of a specific mTEClo subtype (e.g. post-AIRE mTEC) to the mTEClo samples?

How do the authors account for mTEClo being a mixture of cell types, including tuft-like TEC and post-AIRE mTEC? Could this result in different interpretations of any of their findings?

The lack of sequencing saturation seems to be an important limitation in this study. Can the authors be sure that lack of detection is due to genuine absence of a TSR rather than lack of ability to resolve a signal at that locus? This makes the authors' declaration that epitopes are missing worthy of caveat.

The epitope database appears to be password protected at present.

Reviewer #2 (Thymocyte development, transcription regulation) (Remarks to the Author):

NCOMMS-21-42425-T

"Epitope diversity in human medullary thymic epithelial cells"

By Jason A. Carter et al.

Thymic epithelial cells (TEC) control the selection of a T cell repertoire reactive to stranger but tolerant of self. This process (Negative selection) is known to involve the promiscuous gene expression (pGE) of virtually the entire protein-coding gene repertoire, but the extent to which TEC recapitulate peripheral isoforms, and the mechanisms by which they do so, remain largely unknown.

There are several reports to challenge this mystery in mice, however analysis in human is very little or none.

Therefore, this manuscript with Tour de force is good attempt to resolve this mystery.

They had three point to argue about this mystery: mis-initiation, alternative splicing, and expression of ERE. These data set will be the base-line for the future discussion to solve this mystery

I prefer to hear their response to comments (see below) before acceptance.

Comments:

(1) pGE is for the negative selection (e.g. T cell repertoire not reactive to self), however in the case of so-called "agonist selection (for example, generation of tTreg, etc.)" these T cell must be self-reactive to prevent autoimmunity. Is there any difference between pGE for normal developed T cell not reactive to self, and self-reactive tTreg. I would like to see some discussion or data (if possible) on this manuscript.

(2) In Fig6D, it is conceivable that many ERE-promoter driven lncRNA may have regulatory function for the transcription of pGE, is there some evidence about that?

(3) Minor comment: full name of TPA (at page 11) should be appeared somewhere, as lectin *Tetragonolobus purpureas* agglutinin (TPA).

Reviewer #3 (Thymic epithelial cell, autoimmunity, AIRE) (Remarks to the Author):

The manuscript by Carter et al examines in mature and immature human mTECs the transcript diversity associated with mis-initiation and alternative splicing, as well as promiscuous gene expression and ERE enrichment.

The authors found that mature mTECs (mTEChi) have increased rates of transcript mis-initiation in comparison to immature mTECs (mTEClo) and that it may be promoted by AIRE. They confirm in humans the high transcript diversity previously shown in mice, show that it is also a feature of mTEClo and that the induction by AIRE is unlikely to impact the transcript diversity. In addition, they show that pGE is similar in humans and mice, and that ERE expression, although associated with mTEC maturation, may be a consequence of pGE.

This work provides interesting new information on promiscuous gene expression at the gene and transcript levels in human mTECs. The subject area is of significant interest. Conclusions of the paper are well supported by the data with appropriate controls and statistical tools employed. Bioinformatics is well done and full details are available. However, additional evidence to specify the impact of AIRE-induced expression on transcript mis-initiation and alternative splicing is necessary to increase the impact of the study.

1) The fraction of mTEChi- and mTEClo-specific TSRs mapping the known promoters (Fig1D - 2E) and the different genomic locations (Fig 1E) should be shown for the AIRE and the FEZF2-induced TRA genes, but also for the [AIRE and FEZF2]-independent TRAs and non-TRAs, as controls.

2) The extent of transcript mis-initiation should also be specified by showing and comparing the fraction of genes with mis-initiation among the AIRE-dep TRA, FEZF2-dep TRA, [AIRE and FEZF2]-indep TRA and non-TRA genes in mTEChi and in mTEClo. The main question is do the AIRE-dep genes expressed in mTEClo have a lower level of mis-initiation than the same set of genes in mTEChi?

Since TSR identification is dependent on gene expression, comparison of mis-initiation in the different above categories should take gene expression into account as a confounding variable, by instance in considering expression-matched genes.

3) Similarly, selecting the AIRE-dep genes and the above control gene sets in mTEChi and in mTEClo, then performing the splicing entropy (Fig 5A) and alternative splicing analysis (Fig 5D) would help strengthening the conclusion that AIRE induction is neutral to the alternative splicing in mTEChi. This observation would fit with the conclusion of a recent paper studying splicing in mouse (Padonou et al, EMBO Reports 2022).

4) Epitope mapping is challenging in mTECs due to the low number of cells that can be sorted from a human thymus, impairing peptidomics experiments. The authors state in the abstract that this study represents an important step towards the generation of a comprehensive epitope map in the human thymus. However, the main text seems to lack parts explaining how the generated results help defining such a map.

Minor comments:

1) Related to Fig1 A and B: isn't it expected that a higher fraction of mTEChi-specific TSRs is associated with TRA, AIRE and FEZF2-induced genes? Indeed, a fraction of these genes may not be expressed in mTEClo, making their TSRs observed in mTEChi only. If so, the author should indicate in the text that these results were expected.

2) The authors identify more splicing factors upregulated in mTEClo than in mTEChi (Fig 5E). However, this surfeit might not be directly correlated to a higher level of splicing in mTEClo than in mTEChi. Indeed, splicing factors can exert opposite effects on alternative splicing, some activating, others inhibiting or having mixed effects on the inclusion of alternatively spliced exons.

3) Few labels are missing in Fig1 (D, F). In Fig1A pGE and MHCII seem to be missing too. I'm not sure to understand why there is one blue and two purple rectangles in front of the missing MHCII and of AIRE and FEZF2?

We would first like to thank the reviewers for their insightful and constructive comments that have allowed us to significantly improve our manuscript. We provide a detailed point-by-point response in blue for each reviewer below. We have highlighted changes to the updated manuscript in green.

Comments to Reviewers

Reviewer #1 (Thymic selection, systems immunology) (Remarks to the Author):

The authors use 5'Cap- and RNA-seq to profile expression profiles in human thymic samples, with cells purified from the mTEClo and mTEChi compartments. Overall, the authors use this dataset to assess PGE and mTEC maturity in a number of ways. It comprises a useful resource to the community of thymic immunobiologists but I have a number of reservations with regards to the manuscript's conclusions.

Specific comments

Inconsistent capitalisation of genes/protein names plus PGE vs. pGE.

We have carefully revised the display of all gene/protein names for mouse and human, following these conventions:

- Mouse: gene symbols are italicized, with only the first letter in upper-case
- Human: gene symbols are italicized characters that are all in upper-case
- Protein names for both species appear in all uppercase and are not italicized.

Promiscuous gene expression is now abbreviated as pGE throughout the manuscript.

“with minor correlations between samples taken from the same individual” – could the authors explain more about what this means? How often were the same TSRs observed in replicate samples from the same patient? If this happens infrequently, does that mean that some observed signals could be sequencing noise rather than genuine TSRs?

As the reviewer notes, our concern was that sequencing noise and the comparison of mTEClo and mTEChi samples taken from the same individual could confound our comparison of mTEClo and mTEChi populations if there are individual-specific TSRs. To explore this, we calculated the pairwise correlation of TSR expression between all mTEC samples (Supplementary Figure 3A). Reassuringly, we found that the correlation between mTEChi-mTEChi and mTEClo-mTEClo samples from different individuals were higher than the correlation between mTEChi-mTEClo samples taken from the same individual (Supplementary Figure 3B). The sharing of TSRs across mTEC samples taken from different individuals strongly supports the reliability of the observed TSRs rather than sequencing noise.

To clarify this point, we have updated the main text to read:

Results, Lines 93-97: To investigate this, we first needed to understand TSR usage patterns on global and sample level. Globally, principal component analysis on normalized TSR expression

levels across mTECs showed a clear separation of the mTEChi and mTEClo populations (Figure 1F). On sample level, all against all correlation analysis showed that the correlation between TSR expression was greater among mTEChi-mTEChi and mTEClo-mTEClo samples taken from different individuals than the correlation between mTEChi-mTEClo pairs taken from the same individual (Supplementary Fig 3).

Is there any evidence for additional specific TEC chromatin modifications at mTEChi/mTEClo TSRs?

There is some evidence from mouse ChIP-seq demonstrating that the chromatin architecture is established early in mTEC differentiation (Handel et al 2018). Unfortunately, we are unaware of any human ChIP-seq data in mTECs and therefore unable to comment on specific TEC chromatin modifications at mTEClo/hi TSRs.

On a similar note, were the enhancer and super-enhancer definitions used here based on the TEC chromatin landscape (rather than pooled thymus tissue, which would be mostly thymocytes)? This could be lifted over from mouse (e.g. enhancer chromatin modifications from <https://www.frontiersin.org/articles/10.3389/fimmu.2018.02120/full>).

The enhancer and super-enhancer definitions used throughout were downloaded from the [SEdb database](https://academic.oup.com/nar/article/47/D1/D235/5146197) (<https://academic.oup.com/nar/article/47/D1/D235/5146197>), which uses a comprehensive set of super-enhancers identified using a total of 240 tissue and cell types. We felt using human super-enhancers, although not specific to TECs, would be more reliable than trying to lift over mouse super-enhancers. Specifically, although enhancer motifs are conserved across species, previous studies have demonstrated that enhancer loci are not conserved (Chen et al PLOS Computational Biology 2018). As lifting over mouse coordinates depends on loci, we instead chose to use the experimentally validated human enhancers from SEdb albeit not TEC specific. Unfortunately, we are currently unaware of any functional genomic data from human mTECs that would allow for the identification of human TEC-specific super-enhancers. We agree with the reviewer that a better understanding of the human mTEC chromatin landscape will be an important next step, but believe this is beyond the scope of the current work. We have updated the main text to read:

Results, Lines 160-164: While the mTEC chromatin landscape has been explored in mice⁵⁹, functional genomic data and thereby TEC-specific super-enhancers is currently not available for human mTECs. As a baseline to investigate the contribution of SEs to transcript mis-initiation, we used the human super-enhancer database (SEdb; representing super-enhancers across 240 human tissue and cell types⁵⁸), and calculated the genomic distance between each TSR and the nearest known SE.

The authors report enrichment for Nfkb, Nanog, Ascl2 and Prdm10 around mTEChi TSRs. There is limited support for these as important in mTEChi vs. mTEClo from RNA-seq. However, there is often a disconnect between transcriptome and proteome for transcription factors: have

the authors tried to assess these transcription factors in mTEChi vs. mTEClo by FACS or similar methods?

We agree that there are often discrepancies between RNA and protein expression, as has been found across several studies looking at genome-wide correlations (de Sousa Abreu *et al* 2009, Mol Biosystems; Vogel *et al* 2012 Nat Rev Genetics). However, for the subset of differentially expressed genes, more recent work has shown a significantly better correlation with their protein product than non-differentially expressed mRNAs (Koussounadis *et al* 2015, Scientific Reports). While we do not have enough remaining tissue from our samples to verify protein level expression as suggested by the reviewer, the findings by Koussounadis *et al* support our confidence in using the set of differential TF expression for inference on a functional, ie protein level.

We have additionally updated the methods to better reflect the Reviewer's above two points related to our typical- and super-enhancer and transcription factor motifs:

Methods, Lines 436-443: Transcription start region motif and enhancer analyses.

Transcription factor motifs were identified within 200 base pairs around each TSR using HOMER⁵¹ (findMotifsGenome.pl). We included the 400 curated transcription factor motifs using the default HOMER list for both mTEClo and mTEChi specific TSRs independently. The mTEClo:mTEChi odds ratio was then calculated according to the number of TSRs in each population containing a given transcription factor motif. Those transcription factors with a Bonferroni-corrected p-value $p \leq 0.05$ were reported. Super- and typical-enhancer coordinates were downloaded from the human super-enhancer database (SEdb)⁵⁸ (<http://www.licpathway.net/sedb/>), which encompasses a comprehensive set of enhancers drawn from 240 human tissue and cell types. We calculated the linear intra-chromosomal distance in base pairs between each of our TSRs and the nearest super- or typical-enhancer in SEdb.

The definition of mTEChi and mTEClo specific TSRs is based on detection in at least 2 samples of one mTEC subtype and non-detection in the other subtype. Is there a statistical rationale for this? Would this be called as significant by any standard differential analysis approach (e.g. DESeq2)?

When compared with traditional RNAseq data, TSR abundance is more variable across samples (Forrest 2014, Nature). Specifically, even TSRs that are highly expressed in several samples may not be detected in many other samples (please see Supplementary Figure 2B). This variability across samples greatly increases the statistical noise of TSR expression and makes it difficult to identify differentially expressed TSRs using traditional differential gene expression pipelines (DESeq2, Sleuth, etc.).

We have included a supplementary note and updated the Methods to better highlight the rationale for our approach:

Results, Lines 97-102: As these findings demonstrate that TSR patterns within mTEC populations outweigh sample specific TSRs, we next sought a population-specific TSR usage cut-off which would allow us to study TSR usage differences in these two populations on transcript level. We defined mTEChi-specific TSRs as those 1,855 unique TSRs representing 1,500 unique genes that were independently observed in at least two mTEChi samples and not observed in any mTEClo sample (see Methods and Supplementary Note: Population-specific TSR definition). Using the same definition, we identified 2,374 TSRs from 2,011 unique genes that were unique to the mTEClo population.

Discussion, Lines 371-373: Deeper sequencing⁷⁸ and additional human mTEC samples will be necessary to refine the definition of mTEC population specific TSRs and achieve comprehensive coverage for comparison with peripheral TSRs.

Methods, Lines 428-434: The variability of TSR expression across samples, and specifically the high prevalence of TSRs expressed only in a subset of mTEC samples, made identification of mTEC population-specific TSRs difficult using conventional differential gene expression approaches (see Supplementary Note: Population-specific TSR definition). In order to identify mTEChi- and mTEClo-specific TSR sequences that were reproducible (i.e. detected in at least two independent samples), we empirically defined mTEChi-specific TSRs as those TSRs that were expressed in at least two mTEChi and not detected in any mTEClo samples; mTEClo-specific TSRs were called accordingly.

Supplementary Note, 1.1 Population-specific transcription start region definition: When compared with traditional RNA-seq data, 5'Cap-seq transcription start region (TSR) abundance is more variable across samples. Specifically, even TSRs that are highly expressed in several samples may not be detected in many other samples (Supplementary Fig 2). This variability across samples increases the statistical noise of TSR expression and makes it difficult to identify differentially expressed TSRs using traditional differential gene expression pipelines such as DESeq2 or Sleuth.

Consider the following example of a TSR that is expressed in none of the five mTEClo samples (represented by vector [0,0,0,0,0]) and expressed at 100 transcripts per million in $\frac{3}{5}$ mTEChi samples ([100,100,100,0,0]). By paired T-test, this yields a p-value of $p=0.07$, which would not be considered statistically significant at the common α level of $p=0.05$, even before correcting for multiple testing. As such, only a very small number of TSRs would be called as differentially expressed using standard methods and we felt it necessary to seek alternative definitions that would allow us to empirically define TEC population-specific TSRs. Pragmatically, we defined a TSR as specific to either of the mTEC populations if it was expressed in at least two samples from that population and not detected in the other population. This ensures that TEC-specific TSRs are reproducible (expressed in at least two independent samples) without being too restrictive for downstream analysis. A further consideration is that our 5'Cap sequencing approach does not allow for absolute saturation and detection of every TSR from each sample (Supplementary Fig 1B). It is therefore possible that some of the 'TEC-specific' TSRs using our

approach might be expressed in both populations. However, incorrectly calling TSRs as specific to one of the populations when they are actually shared would make the two populations appear more similar to one another in our subsequent analyses. The observed differences between the mTEClo and mTEChi populations identified using this method are therefore likely conservative.

Does the analysis account for multiple samples being taken from the same patient?

Yes, we show paired mTEClo and mTEChi samples taken from the same patient and used paired sample statistical testing where applicable. We have updated the Methods to better reflect this point:

Lines 418-420: Throughout the manuscript, we graphically show paired mTEClo and mTEChi taken from the same individual and report the results of appropriate paired-sample statistical testing whenever possible.

Would further sequencing of these libraries improve overlap between samples (as suggested in Figure S1B)?

The Reviewer is correct in that we were unable to reach sequencing saturation levels for the discovery of unique TSRs within our samples. While increasing saturation would increase the overall number of shared TSRs, it is difficult to determine the degree to which complete saturation would change the frequency with which TSRs are shared across multiple samples (particularly as capture of less frequently expressed TSRs not currently detected may be more likely to be unique to that sample). Unfortunately, this question is difficult to conclusively answer as we do not have remaining libraries or tissue from these rare human mTEC samples for additional sequencing. Future studies using highly-specialized single-molecule hCAGE sequencing or recently published high-throughput adaptations of 5'Cap sequencing (Zhang et al. 2021, Cell Reports Methods) with additional human mTEC will likely be needed to definitively address TSR overlap, see revised discussion:

Discussion, Lines 370-373: However, in comparison to standard RNA-seq in the matched thymic samples or hCAGE data from FANTOM5, we find that we have not reached saturation levels for the detection of TSRs in mTECs. Deeper sequencing⁷⁸ and additional human mTEC samples will be necessary to refine the definition of mTEC population specific TSRs and achieve comprehensive coverage for comparison with peripheral TSRs.

Importantly, the majority of our analyses do not directly depend upon the fraction of the TSR repertoire that are shared across mTEC samples. However, we agree that the lack of sequencing saturation is an important caveat for the definition of mTEChi and mTEClo specific TSRs (see comment above and the new supplementary note), the significance of “missing” epitopes (response regarding “missing epitopes” in the Reviewer’s second to last comment) and the comparison of our 5'Cap sequencing data with the FANTOM5 consortium hCAGE data. Concerning the latter point, we do directly compare TSR repertoire overlap between mTEChi/lo and FANTOM5 peripheral tissues in Figure 2G. While greater TSR saturation in the mTEC

samples would not account for the mTEC TSRs not found in FANTOM, it is likely that a portion of the FANTOM-specific TSRs are actually expressed in the mTEC samples. The leave-one-out analysis (Figure 2H) is normalized to the total number of TSRs in a sample and therefore should be minimally affected by sequencing saturation. We have acknowledged this limitation in the main text:

Results, Lines 134-138: We found that approximately 27% (8,268) of mTEChi and 29% (8,680) of mTEClo TSRs were also found in at least one of these 10 FANTOM5 samples, with 37% (16,078) of our unique mTEC TSRs found in both of our mTEChi and mTEClo populations (Figure 2G), though these estimates are potentially limited by the comparative lack of sequencing saturation in our mTEC samples (Supplementary Fig 1B).

“Aire could be the predominate” -> “Aire could be the [predominant]”
Thank you for spotting this typo, it has now been corrected.

Although it looks convincing that TSRs are located outside the promoter region for AIRE-regulated genes, the plot for FEZF2-regulated genes is much less convincing. Given the amount of noise in the FEZF2 Figure 3E plot, can the authors be sure that there isn't a cell type-specific effect, which may be driven by differential relative contribution of a specific mTEClo subtype (e.g. post-AIRE mTEC) to the mTEClo samples?

As discussed more below, we are unable to differentiate between different mTEC subtypes within the mTEClo population post-hoc. However, differences in the relative contribution of subtypes to the mTEClo population would affect all of our analyses, rather than just the FEZF2 plot in Figure 3F (including the similar AIRE analysis in 3E). We therefore feel like the relative noise in the FEZF2 results is most likely due to the lower number of FEZF2 genes (n=256) relative to AIRE genes (n=3,361) as outlined in the Methods section.

How do the authors account for mTEClo being a mixture of cell types, including tuft-like TEC and post-AIRE mTEC? Could this result in different interpretations of any of their findings?

We are unable to differentiate between various subtypes within our mTEClo population, including tuft-like TECs and post-AIRE mTECs. As we mention in the discussion, future studies may be able to use TPA to distinguish post-AIRE mTEClo cells from immature mTEClos. However, we would expect inclusion of post-AIRE mTEClo cells in the immature mTEClo population would make the mTEClo population appear more similar to the mTEChi population.

The lack of sequencing saturation seems to be an important limitation in this study. Can the authors be sure that lack of detection is due to genuine absence of a TSR rather than lack of ability to resolve a signal at that locus? This makes the authors' declaration that epitopes are missing worthy of caveat.

We agree with the Reviewer that the lack of sequencing saturation represents an important limitation for our study. We did not mean to imply that our current set of mTEC TSRs represents a comprehensive set; human mTECs likely express more TSRs than captured in our data.

We have updated the Discussion to better convey this point:

Discussion, Lines 362-373: Ultimately, we hope that future studies will be able to expand upon these features of thymic transcriptome diversity and lead to the creation of a comprehensive database of the epitopes responsible for inducing thymic tolerance in humans. By comparing such a genome-wide set of thymic epitopes with those encountered in the periphery, missing epitopes responsible for auto-reactive T cell escape or for driving immune recognition of tumor neoantigens could be systematically identified. While the present study represents an important first step, comprehensive sequencing of TSRs, complete isoform coverage, and further studies on regulatory mechanisms for ERE expression in human mTECs is still necessary to achieve this goal.

Specifically, our TSR analysis has shown that 5'Cap sequencing followed by standard Illumina sequencing³⁹⁻⁴¹ is a powerful tool for genome-wide transcription initiation analysis without relying on highly specialized single-molecule sequencing. However, in comparison to standard RNA-seq in the matched thymic samples or hCAGE data from FANTOM5⁴⁶, we find that we have not reached saturation levels for the detection of TSRs in mTECs. Deeper sequencing⁷⁸ and additional human mTEC samples will be necessary to refine the definition of mTEC population specific TSRs and achieve comprehensive coverage for comparison with peripheral TSRs.

The epitope database appears to be password protected at present.

The database can be accessed using these credentials:

Username: reviewer1

Password: Dn6s99BhS2hua7hPWzVg

Reviewer #2 (Thymocyte development, transcription regulation) (Remarks to the Author):

Thymic epithelial cells (TEC) control the selection of a T cell repertoire reactive to stranger but tolerant of self. This process (Negative selection) is known to involve the promiscuous gene expression (pGE) of virtually the entire protein-coding gene repertoire, but the extent to which TECs recapitulate peripheral isoforms, and the mechanisms by which they do so, remain largely unknown. There are several reports to challenge this mystery in mice, however analysis in human is very little or none.

Therefore, this manuscript with Tour de force is a good attempt to resolve this mystery. They had three points to argue about this mystery: mis-initiation, alternative splicing, and expression of ERE. These data set will be the base-line for the future discussion to solve this mystery

I prefer to hear their response to comments (see below) before acceptance.

Comments:

(1) pGE is for the negative selection (e.g. T cell repertoire not reactive to self), however in the case of so-called “agonist selection (for example, generation of tTreg, etc.)” these T cells must be self-reactive to prevent autoimmunity. Is there any difference between pGE for normal developed T cells not reactive to self, and self-reactive tTreg. I would like to see some discussion or data (if possible) on this manuscript.

There is some interesting evidence to suggest that there is a niche for anti-viral plasma cells in the thymic perivascular space (Nunez et al 2016, Science Immunology). To our knowledge, however, there is currently little evidence to suggest the existence of spatial pGE niches tailored to specific T cells subsets. That is, we generally think of each T cell being exposed to the same set of epitopes within the thymus with TCR affinity driving the differentiation of self-reactive tTregs (Owen et al. J Immunol 2019). For example, AIRE knockout mice lead to organ-specific autoimmunity via conventional TCR clonotypes that were preferentially found as Tregs in wild-type mice (Malchow et al Immunity 2016). Deletion of FEZF2 also led to relatively small reductions in the number of tTregs (Takaba et al Cell 2015). Of interest, there is evidence that Tuft cells contribute to the development of Treg subsets (Watanabe et al Nature 2005, Owen et al Nature Immunology 2019) and for the role of self-reactive CD4 T cells to induce mTEC_{lo} differentiation and TRA expression (Lopes et al. eLife 2022). As our study focuses on pGE in the mTEC_{lo} and mTEC_{hi} populations, we are unfortunately unable to delineate any interactions between specific T cell subsets and mTEC pGE.

We have updated the Discussion to include this point:

Discussion, Lines 380-381: Finally, studies investigating the specific roles of mTEC pGE in the development of various T cell subsets, including auto-reactive regulatory T cells (Tregs)⁸⁰, will also be of interest

(2) In Fig6D, it is conceivable that many ERE-promoter driven lncRNA may have regulatory function for the transcription of pGE, is there some evidence about that?

We are not aware of any studies investigating the role of lncRNAs on pGE in mouse or human. Overall, transcription and function of lncRNAs in thymic epithelial cells has not been studied in depth with limited work on their role in viral infection of the thymus (Messias, Sci Report 2020), thymic epithelial tumors (Su 2020, Thoracic Cancer) or their necessity for thymocyte adhesion (Duarte 2021, bioRxiv). Moreover, the ERE-promoter driven lncRNA that we uncovered here have mostly unannotated function (see new Supplementary Data, sheet 'ERE-initiated chimeric transcription events') and we can therefore not assess if there is a baseline for this hypothesis. While this is certainly of interest for future studies, we cannot make an assessment without further functional data.

(3) Minor comment: full name of TPA (at page 11) should appear somewhere, as lectin *Tetragonolobus purpureas* agglutinin (TPA).

Thank you for spotting this omission, we have included this in the manuscript now.

Reviewer #3 (Thymic epithelial cell, autoimmunity, AIRE) (Remarks to the Author):

The manuscript by Carter et al examines in mature and immature human mTECs the transcript diversity associated with mis-initiation and alternative splicing, as well as promiscuous gene expression and ERE enrichment. The authors found that mature mTECs (mTEChi) have increased rates of transcript mis-initiation in comparison to immature mTECs (mTEClo) and that it may be promoted by AIRE. They confirm in humans the high transcript diversity previously shown in mice, show that it is also a feature of mTEClo and that the induction by AIRE is unlikely to impact the transcript diversity. In addition, they show that pGE is similar in humans and mice, and that ERE expression, although associated with mTEC maturation, may be a consequence of pGE.

This work provides interesting new information on promiscuous gene expression at the gene and transcript levels in human mTECs. The subject area is of significant interest. Conclusions of the paper are well supported by the data with appropriate controls and statistical tools employed. Bioinformatics is well done and full details are available. However, additional evidence to specify the impact of AIRE-induced expression on transcript mis-initiation and alternative splicing is necessary to increase the impact of the study.

1) The fraction of mTEChi- and mTEClo-specific TSRs mapping the known promoters (Fig1E - 2D) and the different genomic locations (Fig 2E) should be shown for the AIRE and the FEZF2-induced TRA genes, but also for the [AIRE and FEZF2]-independent TRAs and non-TRAs, as controls.

We have now included a new Supplemental Figure, included below, that includes the requested analyses. Specifically, panels A-B include the full genomic location data (as in Figure 2E) for AIRE and FEZF2 (corresponding to main Figure 3E-F) dependent genes across the two mTEC populations. Panels C-D show both these the frequency of TSRs mapping to known promoters (left, as in Figure 2D) and other genomic locations (right) for [AIRE and FEZF2]-independent (“other”) TRAs and housekeeping genes (i.e. non-TRAs), respectively.

Of particular interest, both [AIRE and FEZF2]-independent TRAs and housekeeping genes demonstrate increased rates of transcript mis-initiation in the mTEChi population, similar to that of AIRE-induced genes. As the true set of human-specific AIRE-induced genes is not currently known, we lift-over AIRE inducible genes from a mouse KO model and a portion of the “AIRE-independent” TRAs may actually be AIRE inducible. Nevertheless, the increased transcript mis-initiation rates in mTEChi housekeeping genes is interesting and likely reflects alternative sources of transcript initiation stochasticity such as chromatin remodeling. Importantly, these results do not change our prior analyses directly comparing AIRE and FEZF2 induced genes and our conclusions regarding the potential role of AIRE, but not FEZF2, in transcript mis-initiation (please additionally see new Supplemental Figure 6 described below).

We have updated the main text to include these points:

Results, Lines 196-204: Beyond these AIRE and FEZF2 induced genes, we additionally found that both housekeeping genes and TRAs not known to be induced by either transcription factor had increased rates of transcript mis-initiation in the mTEChi-specific TSR population relative to the mTEClo population (Supplementary Fig 5C-D and Supplementary Fig 6C-D). Limited by the caveat that human AIRE and FEZF2 induced genes have only been defined by lifting over from the corresponding mouse gene sets, these findings suggest that AIRE is not the sole driver of transcript mis-initiation in mTEChi and other factors are involved, likely including differences in global chromatin remodeling⁶¹. In summary, AIRE-induced genes in the mTEChi population had the highest rates of transcript mis-initiation when compared with AIRE- and FEZF2-independent TRAs and housekeeping genes (Supplementary Fig 6), strongly supporting a direct role for AIRE, but not FEZF2, in promoting transcript mis-initiation.

Supplementary Figure 5. mTEChi- and mTEClo-specific TSR distributions by gene type. (A) Corresponding to the AIRE induced genes in Figure 3E and **(B)** FEZF2 induced genes in Figure 3F, the distribution of genomic location annotations are shown for both the mTEChi- and mTEClo-specific TSRs. **(C)** Genomic location distributions are additionally shown for other TRAs (i.e. TRAs that are not known to be induced by AIRE or FEZF2) and **(D)** housekeeping genes. Boxplots show median values with interquartile ranges. TTS- transcription termination site; UTR- untranslated region.

2) The extent of transcript mis-initiation should also be specified by showing and comparing the fraction of genes with mis-initiation among the AIRE-dep TRA, FEZF2-dep TRA, [AIRE and FEZF2]-indep TRA and non-TRA genes in mTEChi and mTEClo. The main question is do the

AIRE- dep genes expressed in mTEC^{lo} have a lower level of mis-initiation than the same set of genes in mTEC^{hi}?

Since TSR identification is dependent on gene expression, comparison of mis-initiation in the different above categories should take gene expression into account as a confounding variable, by instance in considering expression-matched genes.

As requested by the reviewer, we have now included a new Supplemental Figure that demonstrates the fraction of genes with mis-initiation (defined as ≥ 1 TSR outside of a known promoter region) for AIRE-induced, FEZF2-induced, [AIRE and FEZF2]-independent (“other”), and housekeeping genes that have similar expression levels. Indeed, we find that AIRE-dependent genes expressed in mTEC^{lo} have a lower rate of mis-initiation than the same set of genes in mTEC^{hi}.

Similar to the results discussed above, we again found that AIRE-induced, other TRA, and housekeeping genes have higher rates of transcript mis-initiation in the mTEC^{hi} population when compared to the mTEC^{lo} population. As expected, there was no difference in the rate of transcript mis-initiation for FEZF2-induced genes. As discussed above, the higher rate of transcript mis-initiation in mTEC^{hi} cells for [AIRE and FEZF2]-independent TRAs and housekeeping genes is interesting and suggests mechanisms other than AIRE promoting transcript initiation stochasticity. Notably, AIRE induced genes in the mTEC^{hi} population had the highest rates of transcript mis-initiation among the four populations and strongly supports our conclusion that AIRE, but not FEZF2, promotes mis-initiation. We have updated the Results to read:

Results, Lines 177-181: To confirm these findings, we calculated the fraction of genes with mis-initiated transcripts (at least one TSR mapping outside of a known promoter region). As the identification of TSRs is dependent on gene expression, we limited this analysis to a set of transcripts with similar expression levels in our mTEC bulk RNAseq samples. We indeed found that AIRE, but not FEZF2, induced genes had higher rates of transcript mis-initiation in the mTEC^{hi} relative to the mTEC^{lo} population (Supplementary Fig 6A-B).

Supplemental Figure 6. Fraction of genes with mis-initiated transcripts. A set of genes with similar expression levels was identified according to expression levels in the bulk RNA-sequencing data. We specifically considered those ~10,000 genes with expression

between 10 and 30 transcripts per million (n=9,957, mean \pm standard deviation: 16.7 \pm 5.4). The fraction of these genes with at least one mis-initiated TSR (i.e. mapping to a genomic location outside of a known promoter region) were calculated for **(A)** AIRE induced, **(B)** FEZF2 induced, **(C)** other TRAs (i.e. TRAs that are not known to be induced by AIRE or FEZF2), and **(D)** housekeeping genes. ***p \leq 0.001, *p \leq 0.05 by Paired T-test.

3) Similarly, selecting the AIRE-dep genes and the above control gene sets in mTEChi and in mTEClo, then performing the splicing entropy (Fig 5A) and alternative splicing analysis (Fig 5D) would help strengthening the conclusion that AIRE induction is neutral to the alternative splicing in mTEChi. This observation would fit with the conclusion of a recent paper studying splicing in mouse (Padonou et al, EMBO Reports 2022).

We have repeated the splicing entropy analysis (Fig 5A) and plotted the relationship between the number of expressed genes and transcripts (Fig 5C) for the mTEC populations and GTEx peripheral tissues using AIRE dependent, FEZF2 dependent, other TRAs, and housekeeping genes. Consistent with the referenced study of Padonou and colleagues, we found that AIRE dependent genes had overall lower splicing entropy when compared with housekeeping genes in both mTEC hi and lo cells. We found no statistically significant difference between splicing entropy for AIRE dependent genes across the mTEC hi and lo populations. Finally, we found that the mTEChi population expressed a lower number of transcripts per AIRE dependent gene than expected by linear regression. We have included these analyses as new Supplementary Figure 7 and updated the Results to read:

Results, Lines 266-269: Consistent with recent findings in mice³², we found that the mTEChi population expressed a lower than expected number of transcripts per AIRE dependent gene (Supplementary Fig 7A). In contrast, no such difference was observed for the number of transcripts expressed per FEZF2 dependent, AIRE and FEZF2 independent TRA, or housekeeping gene in the mTEChi population (Supplementary Fig 7B-D).

We have additionally examined the differential splicing (Fig 5D) using AIRE dependent, FEZF2 dependent, other TRA, and housekeeping transcripts as predicted by rMATS. Interestingly, both AIRE and FEZF2 dependent genes had more differentially expressed SE transcripts in the mTEChi population, in contrast to the other TRA and housekeeping genes which had more SE transcripts in the mTEClo population. Similar rates of differentially expressed RI, MXE, A3SS, and A5SS transcripts were observed between the two mTEC populations. We have included this as a new Supplemental Figure 8 and updated the Results to read:

Results, Lines 275-279: When specifically considering AIRE dependent, FEZF2 dependent, AIRE and FEZF2 independent TRA, and housekeeping transcripts, alternative splicing patterns were consistent with the whole transcriptome analysis though higher rates of skipped exons were observed for AIRE and FEZF2 dependent transcripts in the mTEChi population (Supplementary Fig 8). Notably, no specific difference in the distribution of alternative splicing events were observed for AIRE dependent transcripts.

Supplementary Figure 7. Alternative splicing in mTECs by gene type. Splicing entropy and a linear regression fitting the number of expressed transcripts as a function of the number of expressed genes (as in Figure 5A,C, respectively) are shown for **(A)** AIRE dependent, **(B)** FEZF2 dependent, **(C)** other TRAs (i.e. TRAs that are not known to be induced by AIRE or FEZF2), and **(D)** housekeeping genes. For each gene type, the mTEC^{lo} and mTEC^{hi} populations are shown alongside 25 peripheral tissue types from GTEx.

Supplementary Figure 8. Differential splicing by gene type. As in Figure 5D, differential splicing between the mTEC^{hi} and mTEC^{lo} populations as predicted by rMATS is shown for **(A)** AIRE dependent, **(B)** FEZF2 dependent, **(C)** other TRAs (i.e. TRAs that are not known to be induced by AIRE or FEZF2), and **(D)** housekeeping genes. Skipped exons (SE), retained introns (RI), alternative 5' and 3' splice sites (A5SS, A3SS), and mutually exclusive exons (MXE).

4) Epitope mapping is challenging in mTECs due to the low number of cells that can be sorted from a human thymus, impairing peptidomics experiments. The authors state in the abstract that

this study represents an important step towards the generation of a comprehensive epitope map in the human thymus. However, the main text seems to lack parts explaining how the generated results help defining such a map.

We thank the Reviewer for bringing this point to our attention and did not mean to imply that our results define a comprehensive epitope map. Rather, our results represent a first step towards a detailed understanding of the human mTEC transcriptome. We have updated this portion of the abstract, which now reads:

Abstract: Our findings represent an important first step towards the generation of a comprehensive map of transcriptome diversity in the healthy human thymus. Ultimately, a complete map of thymic expression diversity will allow for the identification of epitopes that contribute to the pathogenesis of auto-immunity and that drive immune recognition of tumor antigens.

We have additionally expanded our discussion to further clarify this point:

Discussion, Lines 362-376: Ultimately, we hope that future studies will be able to expand upon these features of thymic transcriptome diversity and lead to the creation of a comprehensive database of the epitopes responsible for inducing thymic tolerance in humans. By comparing such a genome-wide set of thymic epitopes with those encountered in the periphery, missing epitopes responsible for auto-reactive T cell escape or for driving immune recognition of tumor neoantigens could be systematically identified. While the present study represents an important first step, comprehensive sequencing of TSRs, complete isoform coverage, and further studies on regulatory mechanisms for ERE expression in human mTECs is still necessary to achieve this goal.

Specifically, our TSR analysis has shown that 5'Cap sequencing followed by standard Illumina sequencing³⁹⁻⁴¹ is a powerful tool for genome-wide transcription initiation analysis without relying on highly specialized single-molecule sequencing. However, in comparison to standard RNA-seq in the matched thymic samples or hCAGE data from FANTOM5⁴⁶, we find that we have not reached saturation levels for the detection of TSRs in mTECs. Deeper sequencing⁷⁸ and additional human mTEC samples will be necessary to refine the definition of mTEC population specific TSRs and achieve comprehensive coverage for comparison with peripheral TSRs. On the transcript level, enhanced epitope maps in mTECs could be generated using long-read sequencing for the discovery of potentially novel transcript isoforms. Peptidomic experiments in human mTECs will additionally be necessary to correlate this transcriptomic epitope expression with the peptide diversity directly observed by developing T cells.

Discussion, Lines 382-386: In conclusion, our study has demonstrated several mechanisms that underlie the generation of transcriptomic diversity in human mTECs. Our results represent an important first step towards the generation of a detailed understanding of the mTEC transcriptome and ultimately the identification of epitopes not seen by developing T cells during the induction of central tolerance. Future comprehensive identification of these missing

antigens will play a crucial role in the identification of epitopes with the potential to trigger auto-immune responses against healthy tissue or drive immune recognition of tumor neoantigens.

Minor comments:

1) Related to Fig1 A and B: isn't it expected that a higher fraction of mTEChi-specific TSRs is associated with TRA, AIRE and FEZF2-induced genes? Indeed, a fraction of these genes may not be expressed in mTEClo, making their TSRs observed in mTEChi only. If so, the author should indicate in the text that these results were expected.

The Reviewer is correct and we expect higher rates of TSRs mapping to TRA, AIRE induced, and FEZF2 induced genes in the mTEChi population due to the nature of pGE. We have now updated the relevant portions of the Results section to clarify this point for Figure 2A-B.

2) The authors identify more splicing factors upregulated in mTEClo than in mTEChi (Fig 5E). However, this surfeit might not be directly correlated to a higher level of splicing in mTEClo than in mTEChi. Indeed, splicing factors can exert opposite effects on alternative splicing, some activating, others inhibiting or having mixed effects on the inclusion of alternatively spliced exons.

We thank the Reviewer for bringing this to our attention and have now removed this comment.

3) Few labels are missing in Fig1 (D, F). In Fig1A pGE and MHCII seem to be missing too. I'm not sure to understand why there is one blue and two purple rectangles in front of the missing MHCII and of AIRE and FEZF2?

The figure appears to have been rendered incorrectly when uploaded for submission. We believe this has been corrected now; the correct version of the figure is shown below:

REVIEWERS' COMMENTS

Reviewer #1 (Remarks to the Author):

The authors have responded in a constructive way and addressed all points raised by this reviewer. As there is no remaining material of the initially created library, it would be unfeasible to request entirely new experiments,ents to address one of the points highlighted, namely sequencing to saturation.

Reviewer #2 (Remarks to the Author):

This revised version is now level for the acceptance.

Reviewer #3 (Remarks to the Author):

The authors have addressed all my concerns and substantially clarified key points by performing the required additional analyses. Congratulations for this hard work.

I would eventually recommend to include the new Sup Fig 6 (or equivalent) into the corresponding main Fig, so the readers directly have the important information that TSR mis-initiation is not restricted to Aire-induced TRA gene

Response to reviewers

We thank the reviewers for their renewed effort in reviewing our manuscript. We addressed the last suggestion from Reviewer 3 (see in below).

Reviewer #1 (Remarks to the Author):

The authors have responded in a constructive way and addressed all points raised by this reviewer. As there is no remaining material of the initially created library, it would be unfeasible to request entirely new experiments,ents to address one of the points highlighted, namely sequencing to saturation.

Reviewer #2 (Remarks to the Author):

This revised version is now level for the acceptance.

Reviewer #3 (Remarks to the Author):

The authors have addressed all my concerns and substantially clarified key points by performing the required additional analyses. Congratulations for this hard work.

I would eventually recommend to include the new Sup Fig 6 (or equivalent) into the corresponding main Fig, so the readers directly have the important information that TSR mis-initiation is not restricted to Aire-induced TRA genes.

We have included the new analyses on promoter usage across different gene sets in main figure 3.